

# Autoencoder-based feature extraction for the automatic detection of snow avalanches in seismic data

Andri Simeon[1], Cristina Pérez Guillén[1], Michele Volpi[2], Christine Seupel[1], and Alec van Herwijnen[1]

[1]WSL Institute for Snow and Avalanche Research SLF
[2]Swiss Data Science Center, ETH Zurich and EPFL, Switzerland

**Correspondence:** Andri Simeon (andri.simeon@slf.ch)

**Abstract.** Monitoring snow avalanche activity is essential for operational avalanche forecasting and the successful implementation of mitigation measures to ensure safety in mountain regions. To facilitate and automate the monitoring process, avalanche detection systems equipped with seismic sensors can provide a cost-effective solution. Still, automatically differentiating avalanche signals from other sources in seismic data remains challenging, mainly due to the complexity of seismic signals generated by avalanches, the complex signal transmission through the ground, the relatively rare occurrence of avalanches, and the presence of multiple sources in the continuous seismic data. One approach to automate avalanche detection is by applying machine learning methods. So far, research in this area has mainly focused on extracting standard domain-specific signal attributes in the time and frequency domains as input features for statistical models. In this study, we propose a novel application of deep learning autoencoder models for the automatic and unsupervised extraction of features from seismic recordings. These new features are then fed into classifiers for discriminating snow avalanches. To this end, we trained three Random forest classifiers based on different feature extraction approaches. The first set of 32 features was automatically extracted from the time-series signals by an autoencoder consisting of convolutional layers and a recurrent long short-term memory unit. The second autoencoder applies a series of fully connected layers to extract 16 features from the spectrum of the signals. As a benchmark, a third random forest was trained with typical waveform, spectral and spectrogram attributes used to discriminate seismic events. We extracted all these features from 10-second windows of the seismograms recorded with an array of five seismometers installed in an avalanche test site located above Davos, Switzerland. The database used to train and test the models contained 84 avalanches and 828 noise (unrelated to avalanches) events recorded during the winter seasons of 2020-2021 and 2021-2022. Finally, we assessed the performance of each classifier, compared the results, and proposed different aggregation methods to improve the predictive performance of the developed seismic detection algorithms. The classifiers achieved an avalanche f1-score of 0.61 (seismic attributes), 0.49 (temporal autoencoder) and 0.60 (spectral autoencoder) and avalanche recall of 0.68, 0.71 and 0.71, respectively. Overall, the macro f1-score ranged from 0.70 (temporal autoencoder) to 0.78 (seismic attributes). After applying a post-processing step to event-based predictions, the avalanche recall of the three models significantly increased, reaching values between 0.82 and 0.91. The developed approach could be potentially used as an operational, near-real-time avalanche detection system. Yet, the relatively high number of false alarms still needs further implementation of the current automated seismic classification algorithms to be used as unique methods to detect avalanches effectively.





## 1 Introduction

Every winter, snow-covered mountainous regions worldwide are exposed to the destructive potential of snow avalanches, causing fatalities and damage to infrastructure. On average in Switzerland, 25 avalanche fatalities occur every winter (Techel et al.,
2016). The catastrophic winter of 1999 resulted in infrastructural damage costing several hundred million Swiss francs (Bründl et al., 2004). Such periods underscored the need for ongoing investments in avalanche prevention measures and providing accurate avalanche forecasts. Avalanche forecasting is mainly driven by analysing weather measurements and forecasts in combination with snowpack and avalanche observations (Schweizer et al., 2020). Detailed information on the location and timing of avalanche occurrences is indispensable for validating avalanche forecasts (e.g. van Herwijnen et al., 2016; Bühler
et al., 2022), effectively implementing mitigation measures (e.g. McClung and Schaerer, 2006; Alec van Herwijnen and Techel, 2018), hazard mapping (e.g. Bühler et al., 2022) and the development of statistical approaches to predict natural avalanche release (Sielenou et al., 2021; Hendrick et al., 2023; Mayer et al., 2023). However, avalanche activity data are still mainly obtained through human field observations, which are especially incomplete and uncertain in poor visibility conditions during storms when avalanche activity is usually high (Schweizer et al., 2020). Hence, the demand for automated avalanche detection
systems that provide reliable and continuous avalanche activity data is rapidly growing.

Since avalanches are extended moving sources of seismic energy, seismic monitoring systems can be used to detect natural avalanches in large areas within a radius of several kilometres (Hammer et al., 2017; Pérez-Guillén et al., 2019; Heck et al., 2019b), regardless of the weather and visibility conditions. Seismic avalanche detection systems have been employed for several decades to monitor and characterise avalanches (Suriñach et al., 2001; Biescas et al., 2003; van Herwijnen and Schweizer,
2011), assess the source location (Lacroix et al., 2012; Pérez-Guillén et al., 2019; Heck et al., 2019a) and infer flow properties (Vilajosana et al., 2007; Lacroix et al., 2012; Pérez-Guillén et al., 2016). Avalanches generate spindle-shaped, high-frequency signals similar to other types of mass movements (Suriñach et al., 2005). These patterns have frequently been used to discriminate avalanche signals from other seismic sources. Nevertheless, seismic detection systems have not yet reached the same level of reliability compared to other systems, such as radars, when it comes to the automatic detection of avalanches (Schimmel
et al., 2017). This limitation is partly due to the complex signal transmission from the source (i.e., the avalanche) to the receiver and multiple sources of environmental noise (e.g., earthquakes, aeroplanes, etc.).

The first attempt to automatically distinguish avalanches from other sources based on seismic features observed in the time-frequency domain and combined with fuzzy logic was conducted by Leprettre et al. (1996). Afterwards, Bessason et al. (2007) developed a nearest-neighbour approach that successfully detected 65% of previously confirmed avalanche events. Later, Rubin
et al. (2012) divided a seismic data stream into 5 s time windows and extracted 10 spectral features by applying a Fast Fourier Transform (FFT). Several machine-learning classifiers were tested using these input features, such as random forest algorithms, support vector machines, and artificial neural networks. The decision stump classifier reached the highest precision of 13.2% on manually identified avalanches, while they reported a recall of 89.5% and an accuracy of 93.0%. More recently, Hammer et al. (2017) and Heck et al. (2018) applied hidden Markov models (HMMs) to learn class characteristic patterns based on





extracted spectral features for automatic avalanche classification. Extending on this approach, Heck et al. (2019a) trained an HMM-based method to detect avalanches in continuous seismic data.

In recent years, the extensive growth of collected data and the emergence of machine learning algorithms have opened up new perspectives for efficient and automated data processing. Machine learning models can handle complex data sets in a reasonable time and rapidly synthesise data processes, providing valuable and complementary insights into data (Mousavi and

Beroza, 2022). Over the past decade, statistical and machine learning methods have been developed for automatically classifying seismic signals generated by different types of slope failures based on Hidden Markov Models (Hammer et al., 2013; Dammeier et al., 2016), fuzzy logic (Hibert et al., 2014) and Random Forest algorithms (Provost et al., 2017). So far, these approaches relied on carefully engineered features derived from processing signals in the time and frequency domains. In contrast, we explored a novel approach to automatic feature extraction by developing two unsupervised autoencoders for learning

temporal and spectral signals. Autoencoders, introduced by Rumelhart et al., are neural networks specialised in extracting relevant features from the data, relying on unsupervised learning. They can be directly trained on raw input signals and thus are not dependent on labels or expert-based tuning of specialised feature extractors. The architecture traditionally consists of an encoder and decoder, where the former embeds an input signal to a bottleneck layer, i.e. the latent space, where relevant information is stored. This low-dimensional embedding is designed and trained to include all the relevant information of a

given signal in a feature vector of lower dimensionality than the input. For example, Mousavi et al. (2019) used an autoencoder to cluster seismic signals of an earthquake catalogue and showed comparable precision to supervised methods. Kong et al. (2021) evaluated different autoencoder architectures for seismic event discrimination and phase picking.

In this study, we explored the autoencoder model for automatic feature extraction from seismic signals generated by avalanches and other sources. First, we compiled a catalogue of seismic events recorded at our study site above Davos (Sect.

2), Switzerland, throughout the winter seasons of 2020-2021 and 2021-2022. In Sect. 3, we described the foundation of this dataset, which is one of the most critical parts of any machine learning model development. Similar to previous studies, we extracted features from 10 s seismic time windows and trained classifiers based on these features. In the feature extraction process (Sect. 4.1), we implemented two new methods based on autoencoders, automatically tuned to extract 32 and 16 input features, and compared them against a set of 57 standard expert-based seismic attributes. The routines to optimize and train

the autoencoder models is shown in Sect. 4.2. Using the different sets of input features, we trained three random forest classifiers to automatically distinguish the avalanche signals from other seismic events (Sect. 4.3). We analyzed and compared the performance of the models in Sect. 5. Finally, a discussion of the main results and conclusions is presented in Sect. 6 and 7.

## 2 Study Site

The study site is located at the end of the Dischma Valley, a tributary valley above Davos, Switzerland (Fig. 1). The seis-

mic system was deployed on a flat meadow at about 2000 m a.s.l. (Eastern Swiss Alps; 46.72°N, 9.92°E). The surrounding mountains form a basin of steep slopes reaching up to 3000 m a.s.l. Since the winter season of 2020-2021, approximately from November to May, we installed a seismo-acoustic array of five co-located seismic and infrasound sensors arranged in a





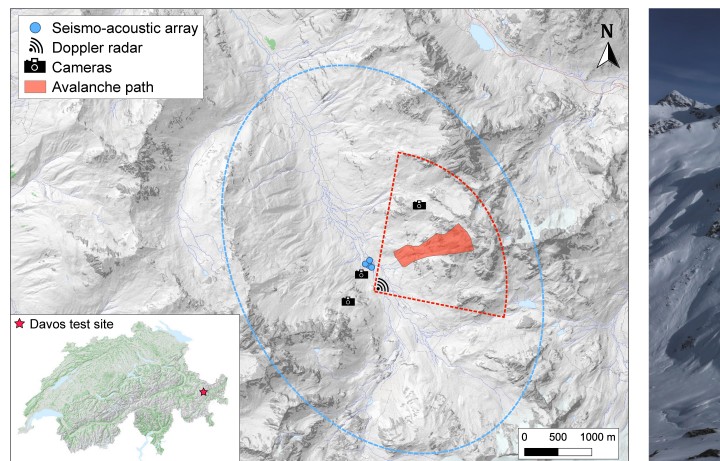 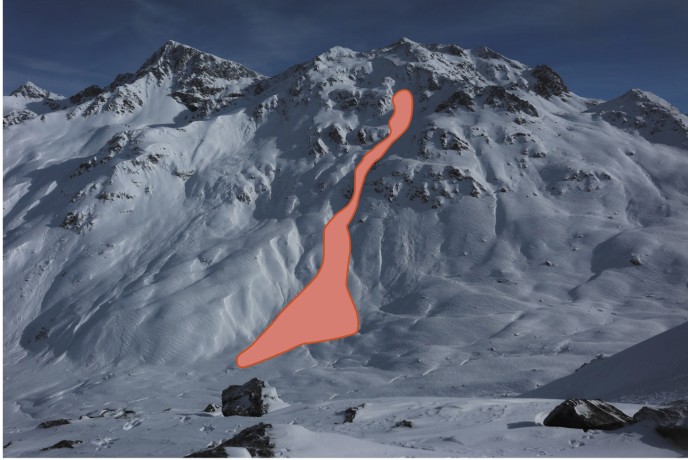

**Figure 1.** Right: Map and location of the study site. The instrumentation consisted of a seismo-acoustic array (blue dots), three cameras and a Doppler radar (red triangle). The approximate area where avalanches can be detected is shown for the seismo-acoustic array (blue ellipse) and the radar (red cone). Moreover, an avalanche path is highlighted with the red shaded area. Left: Photo taken by an automatic camera at the Dischma study site, showing the georeferenced path of a dry-snow avalanche released on 2 February 2022 at 02:31.

star-like pattern. The seismic sensors were buried into the ground at a depth of approximately 50 cm and subsequently covered by snow during winter. A single measuring unit consists of a one-component seismometer Lennartz LE-1D/V (eigenfrequency of 1 Hz and sensitivity of $800\,\mathrm{V\,m^{-1}\,s}$) and an infrasound sensor Item-prs (frequency response of 0.2-100 Hz and sensitivity of $400\,\mathrm{mV\,Pa^{-1}}$). The central seismic sensor consisted of a three-component seismometer LE-3Dlite (eigenfrequency of 1 Hz and sensitivity of $800\,\mathrm{V\,m^{-1}\,s}$), of which we only used the vertical component in this study. The sensors were connected to the same digitizer (Centaur digitizer from Nanometrics), recording continuously with a sampling frequency of 200 Hz. The seismo-acoustic array monitors avalanches released from all slopes within a radius of approximately 3 km (blue ellipse in Fig. 1).

Additionally, the site is equipped with a Doppler radar and three automatic cameras to obtain independent validation data, including accurate release times and information on the type and size of the avalanches. The radar emits electromagnetic waves that are reflected by the avalanche flow, providing the location and velocity of the moving avalanche (Meier et al., 2016). Figure 1 shows the location of the radar, which monitors several avalanche paths exposed to the west-southwest, covering an approximate area of $4\,\mathrm{km^2}$ (red delineated area in Fig. 1). In this case, avalanches can be detected up to a maximum distance of approximately 2 km. The cameras automatically photograph every 30 minutes all the surrounding slopes (Fig. 1).





## 3 Data

We compiled a catalogue of seismic events from the continuous recordings of the winter seasons 2020-2021 and 2021-2022. Concretely, we manually picked events within periods of known avalanche activity and pre-processed the seismic signals. Then, three experts labelled the events, with which we compiled a two-class classification dataset.

### 3.1 Event picking and signal processing

Supervised machine-learning models require a definition of events and a subsequent data annotation for training. For the former requirement, we picked events from the continuous recordings. Typically, the amplitude of seismic signals generated by avalanches gradually increases since the avalanche approaches the location of the seismic sensors (Fig. 1) and larger seismic energy dissipation due to snow entertainment and erosion processes within the flowing avalanche (Pérez-Guillén et al., 2016). As avalanche signals gradually emerge from background noise and initially have a low signal-to-noise ratio (Fig. 2a), automated picking methods often miss the starting phase of avalanches and sometimes entire events. To prevent this, we visually inspected the continuous seismic recordings and identified signals that exhibited a high signal-to-noise ratio, i.e. were not in the order of magnitude of the background noise. We limited our search to periods with known avalanche activity for efficiency. This included avalanche cycles during snow storms, days when avalanches were detected by the radar and periods with observed avalanche deposits in the cameras.

Before picking the signals in those periods, we transformed the raw seismic signals from the five sensors to ground velocity (meters per second). Additionally, the signals were linearly detrended, tapered with a Hanning window and filtered with a 4th-order Butterworth band-pass filter between 1 and 10 Hz. We found this to be the most energetic frequency band of the avalanche signals recorded at our study site, considering the typical relative distance between the avalanche and our receivers. Finally, we manually cut the identified signals to compile a clean event catalogue. To manually pick the start and end times, we visually inspected the seismic signal, the envelope signal and the spectrogram. In summary, we picked 912 non-background noise signals lasting between 5 and 515 s, which we labeled in the next step.

### 3.2 Labelling of events

For the annotation and labelling of events, three experts assigned signals into two classes, avalanches and non-avalanche events:

**Avalanches:** Avalanche events were first identified using the radar and camera data (Fig. 1). We did this by matching seismic signals to the avalanches observed in the radar data or on images. A second step to collect avalanches missed by these systems was to visually classify signals based on the characteristic seismic signature of avalanches (e.g. non-impulsive onsets, spindle-shaped signals and triangular-shaped spectrograms; Fig. 2a) as proposed by van Herwijnen and Schweizer (2011). Additionally, the output of wave parameters derived from array processing of the seismic and infrasound data was considered, i.e. backazimuth angles and apparent velocity (Marchetti et al., 2015; Heck et al., 2019a).



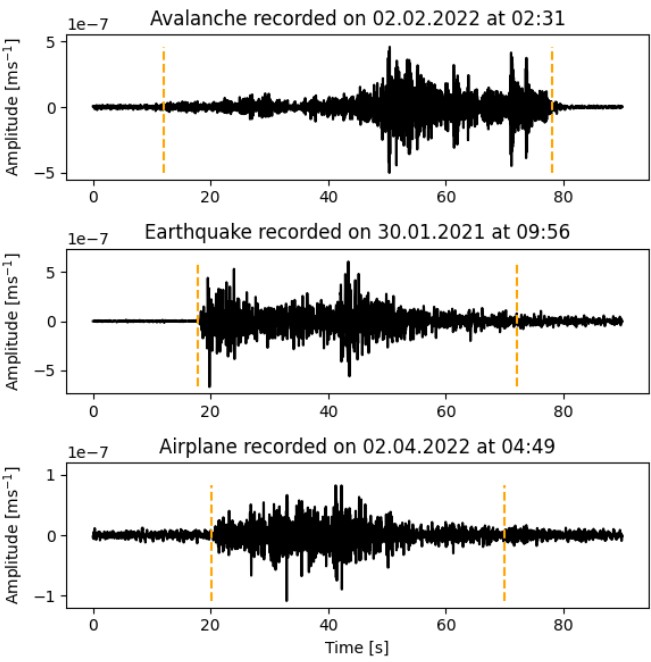

**Figure 2.** Recordings of the avalanche in Fig. 1, an earthquake and an airplane. The dashed orange vertical lines indicate the manual cuts of the event catalogue.

**Noise (non-avalanche events):** Earthquakes were the most frequent source of environmental noise at our study site. They were identified by visual inspection of the signals (typical emergent onsets and usually identifiable arrival of the different phases; Fig. 2b) and comparison of our seismo-acoustic recordings with two nearby seismic stations from the Swiss
National Network (e.g. Clinton et al., 2011). In addition, online earthquake catalogues were consulted to match our recordings with catalogued events (SED, 2023; EMS, 2023). The remaining portion of seismic events was generated by different sources, including aeroplanes (Fig. 2c), helicopters, explosions in nearby skiing resorts, weather events (e.g. wind), people or animals walking close to the sensors, and many more unknown event sources. We summarized this collection of unrelated events as "noise" class. In particular, weak signals generated by non-verified small avalanches
might also fall into this heterogeneous noise class. Notably, this definition of the noise class barely included low SNR background noise.

To label avalanche events, three experts independently assigned subjective probabilities using either 0 (unidentified avalanche), 0.5 (potential avalanche) or 1 (certain avalanche). A signal was labelled positive if the sum of the three expert scores exceeded 1.5. Note that the average rate of agreement in avalanche score on the avalanche signals between the three experts was 58%. In
this manner, we compiled an event catalogue with 84 avalanches (31 verified with the radar or camera images) and 828 unrelated noise events from the 2020-2021 and 2021-2022 winter seasons. For completeness, the same labelling process was used for earthquakes, with which we found 183 earthquakes in the noise class. The seismic sensors recorded maximum absolute





amplitudes ranging from $3.3 \times 10^{-8}$ to $4.7 \times 10^{-5}\,\mathrm{m\,s^{-1}}$ for avalanches, $1.3 \times 10^{-8}$ to $9.7 \times 10^{-6}\,\mathrm{m\,s^{-1}}$ for earthquakes and $1.4 \times 10^{-9}$ to $5.1 \times 10^{-5}\,\mathrm{m\,s^{-1}}$ for noise signals. Signal duration ranged from 13 to 113 s, 7 to 263 s and 5 to 515 s in each
class, respectively. Noteworthy, the amplitude range of the noise class includes the amplitude ranges of both avalanches and earthquakes, highlighting its heterogeneity.

### 3.3   Signal windowing and dataset splitting

Before training the models, we further processed the event data in the catalogue. First, we treated the records of each seismic sensor independently yielding a five-fold enlargement. Second, we applied a 10 s windowing with 50% overlap to all signals.
This windowing resulted in more data samples to train, as the models only received a fixed-sized input. We note that this strategy might also be beneficial in a potential (near) real-time detection system, where 10 s windows are continuously parsed. With this, the labelled data set comprised 3'580 avalanche and 37'110 noise (non-avalanche) windows, which include 11'575 earthquake windows. This dataset is at the core of this study and allows us to systematically compare the methods in different settings.

Lastly, to learn the model and select the best architectures and hyper-parameters, we defined four independent data folds, i.e. three train folds for cross-validation and a test fold for assessing the error on an independent inference set. We separated the folds by specific dates to not induce any correlation between the folds and reduce temporal data leakage. We chose the dates such that the class distributions across the folds are even (Fig. 3). The first train fold included dry avalanches exclusively, whereas the second contained a mixture of dry avalanches in the early part of the period, and wet avalanches in the latter.
The third train fold and the fourth test fold spanned the winter season of 2021-2022. Again, the earlier counted towards dry conditions and the last both wet and dry.

## 4   Model development

In order to classify each signal window (Fig. 4), we need to extract features from them (Sect. 4.1), followed by a classification model learned to discriminate classes of interest (Sect. 4.3). In the former, we used a conventional human-supervised feature-
engineering approach (Sect. 4.1.1 and Appendix B) as a benchmark and two fully unsupervised autoencoders (Sect. 4.1.2), which required definitions of the training strategies (Sect. 4.2). In the latter, we chose and developed binary classifiers for the preceding feature extraction methods (Sect. 4.3).

### 4.1   Feature extraction

Feature extraction generally describes the compression of a signal to a lower dimensional embedding while retrieving/pre-
serving the signal's most distinctive information. The embedded information (the features) is usually input into an upstream classification or regression task. Following this general approach, we explore three methods to extract information from seismic signals either as lower dimensional feature vectors or domain-specific features, which are then classified as avalanche or noise.



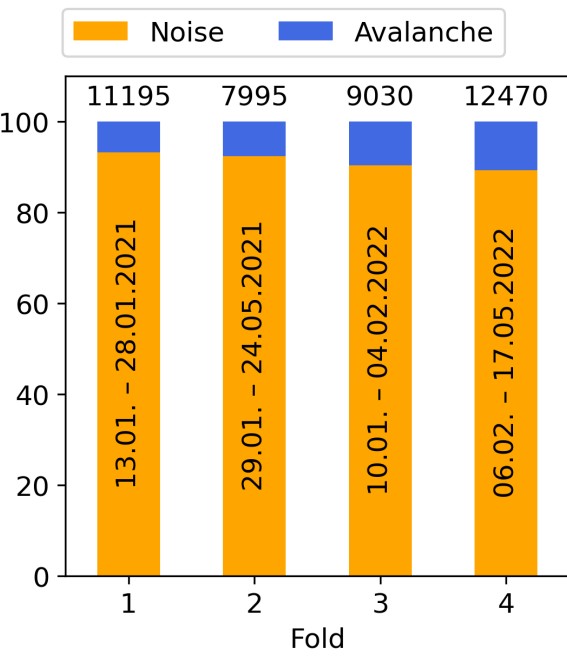

**Figure 3.** Class distributions in the folds. The annotations on top of the bars depict the total number of 10 s seismic windows in each fold.

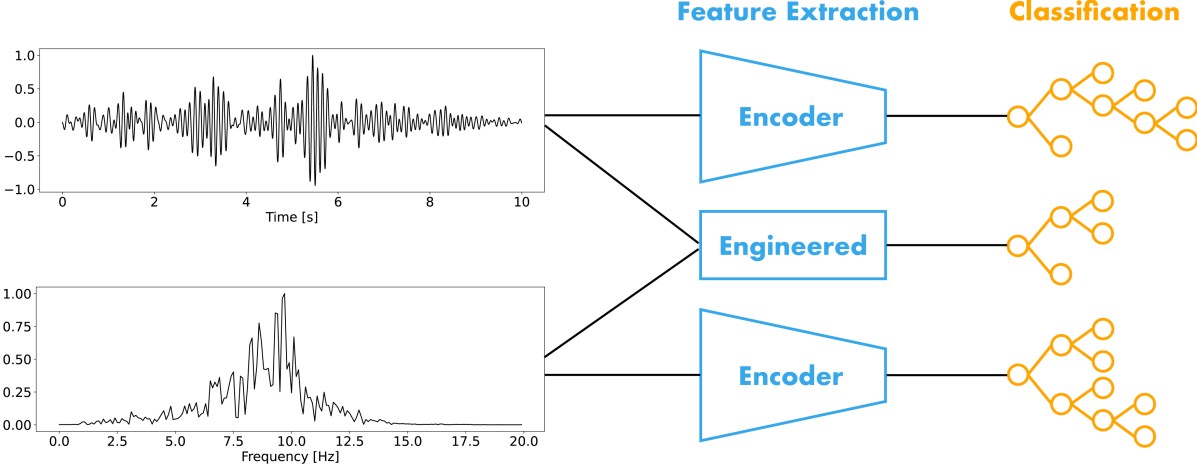

**Figure 4.** Overview of the three different approaches for avalanche classification. The blue elements depict the feature extraction, while the orange parts show the classification. Top (blue): The temporal autoencoder features; middle: The hand-engineered seismic attributes; bottom: The spectral autoencoder features.

In a first attempt, following a similar approach to Provost et al. (2017), which classified seismic events generated by landslides, we extracted a set of 57 predefined standard seismic attributes (Sect. 4.1.1). The feature engineering approach is widely





used in seismic detection of mass movements (Rubin et al., 2012; Provost et al., 2017; Lin et al., 2020; Wenner et al., 2021; Chmiel et al., 2021) and time series classification in general (Barandas et al., 2020). Additionally, it served as a benchmark for comparing our second approach (Sect. 4.1.2), which is to learn the feature extraction completely unsupervised without making any preliminary assumptions about the signals. Using an unsupervised approach is beneficial when not having ground-truth labels, as in our case. Therefore, we used two autoencoder models to extract features from temporal and spectral input data,

respectively (Sect. 4.1.2). The autoencoder architecture, which was first introduced in Rumelhart et al., and has since been adapted for various applications (Lu et al., 2013; Mousavi et al., 2019; Gu et al., 2021). A vanilla autoencoder consists of an encoder and a decoder: The encoder compresses the input signal to a lower-dimensional embedding, the latent space. The decoder transforms feature vectors from this latent space to the original input dimension. An autoencoder is trained by learning to reconstruct the input signals from the lower-dimensional latent space, which requires the latter to store the most relevant

information characterising each piece of the signal. By design, the feature vectors are optimized to carry the most distinctive information of a given input signal, such that the decoder can reconstruct it. During inference, the decoder is discarded, and only latent vectors are used as inputs to the classifier, which is trained separately.

### 4.1.1 Seismic attributes

In the first approach, we used a set of 22 waveform, 17 spectral and 18 spectrogram attributes (see Table B1, B2 and B3 for

more details). These features were extracted from the frequency-filtered (1 to 10 Hz) and normalized 10 s seismic signals for all sensors separately. Note that we did not include any network or polarity-related attributes.

### 4.1.2 Autoenconders

Developing neural networks involves optimizing network hyper-parameters and defining a training strategy. Therefore, we used the first three folds in Fig. 3 to run 3-fold cross-validation. We defined a grid of hyper-parameter combinations, iteratively

trained the models on two and evaluated them on the left-out fold. We selected the model showing the best average performance on all three folds according to predefined metrics. By definition, the autoencoder performance can be measured with its reconstruction loss. However, given a decent reconstruction, we aimed to find the best input features for classification. Hence, we evaluated the autoencoders based on the avalanche and noise class separation within the latent (feature) space. We calculated the silhouette score (Rousseeuw, 1987) and the Calinski-Harabasz index (Caliński and Harabasz, 1974) for the features and

their given expert labels (see Appendix C). The best autoencoder was selected by searching for the highest-ranking combination of silhouette score, Calinski-Harabasz index and the reconstruction mean squared error loss (see Appendix F). Following the model selection, the autoencoders were retrained on the train folds (fold 1, 2 and 3 in Fig. 3), and after, we extracted the autoencoder features from all folds.

In the first autoencoder, i.e. the temporal autoencoder (TAE), we considered the seismic time series data, hence the name.

It was developed for seismic waveform signals of 10 s normalized by their absolute maximum amplitude. When dealing with time series data, common choices of computational units are one-dimensional convolutions and recurrent units such as the long short-term memory (LSTM) cells. Thus, we implemented the encoder as a sequence of 3 convolution layers and one





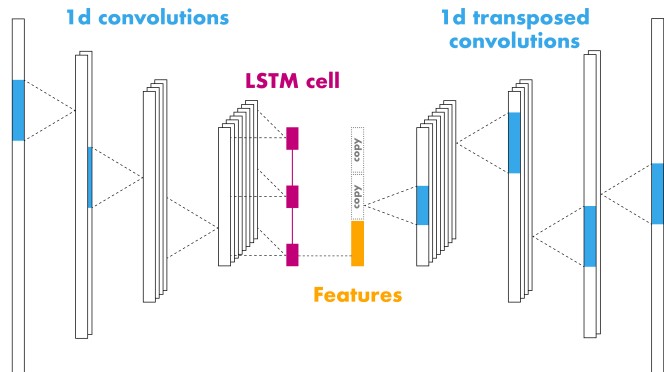

**Figure 5.** Illustration of the architecture of the temporal autoencoder.

LSTM cell layer learning temporal dynamics. The best model from the cross-validation procedure (Table F2) was composed of convolutions with kernel size 20 (or 0.1 s) and stride 10. This implementation of stride reduces the initial input length of 2000 samples (200 Hz × 10 s) to 200, 20, and 2 within each encoder layer. Similarly, we selected 32 filters in the first convolutional layer and doubled the number in each consecutive layer. In the last encoder layer, the LSTM cell summarizes the output of the convolutions, i.e. two 128-dimensional vectors, to a feature vector of 32 dimensions (32 features). The decoder sequentially repeats this latent vector twice and applies 3 transposed convolutions with kernel size 20 and stride 10 to decompress the sequence back to its original length. Starting at 128 filters, we halved them in each layer to reach 32 channels. To reduce this number back to the number of input channels, i.e. 1, a convolutional layer with kernel size 3, stride 1 is applied in the decoder output layer.

In addition, we used batch normalization (BN) (Ioffe and Szegedy, 2015) in all encoder and decoder layers except for the decoder output layer to stabilize and accelerate training. As an activation function, we use the leaky rectified linear unit (leaky ReLU; (Xu et al., 2015)). The only exception is again the output layer, where we replace the leaky ReLU with the tangent hyperbolic (Tanh) function to output values in the same range as the normalized input signals in $[-1, 1]$. In summary, Fig. 5 gives a simplified overview of this architecture comprising 514'337 learnable parameters (226'848 in the encoder). Note that this architecture is relatively small in the number of trainable parameters, hence well adapted to the size of our dataset.

The second autoencoder implementation operates in the spectral domain, henceforth referred to as the spectral autoencoder (SAE). We used the fast Fourier transform (FFT) to convert the filtered 10 s seismic signals into the frequency domain. Thus, the input data to this model contains the amplitude spectrum normalised using the min-max normalization. In contrast to the temporal autoencoder, we replaced the aforementioned computational units, i.e. convolutions and LSTM cells, with fully connected layers. Through hyper-parameter optimization, we designed the encoder and decoder to compose 3 fully connected linear layers. The hidden dimensions in the encoder evolve from 200 to 139, 78 and 16 (feature dimension). The decoder is a mirrored version of the encoder. We used the Tanh function as the non-linearity of choice in all layers (Table F3). Moreover, we apply layer normalization (LN) in each layer with the same exception of the output layer. Figure 6 illustrates a simplified





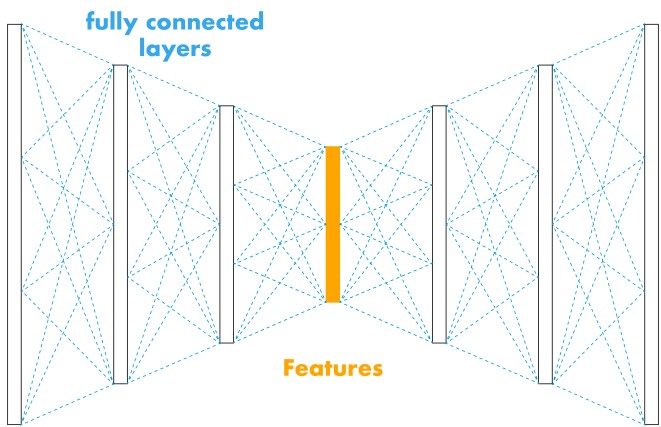

**Figure 6.** Illustration of the architecture of the spectral autoencoder.

version of this architecture summing up to 81'330 learnable weights (40'589 in the encoder). As for the TAE, this architecture is small and well adapted to our dataset.

### 4.2 Autoencoder training

The training strategy is another main part of model development, which we optimized for the selected autoencoder architec-
245 tures. A training step in neural network optimization starts with sampling a batch of predefined size from the dataset. For sampling, given that the data set is severely imbalanced (Fig. 3), we implemented the weighted random sampler (as implemented in Paszke et al. (2019), see Appendix D), which samples data points according to user-specified class weights. This allowed us to control the proportion of avalanche samples within each batch. The batch is then passed through the entire network (forward pass) to produce the output (prediction). The output is compared to the target and the reconstruction loss (Mean
250 Squared Error – MSE) is computed. The network weights are then optimized by computing the gradients of the loss function and applying a specified back-propagation algorithm. Within this training procedure, we searched for the optimal number of expected avalanche samples in each batch, the batch size and the learning rate to use with the Adam optimizer (Kingma and Ba, 2014). After following our hyper-parameter optimization strategy, we found the temporal autoencoder optimal with an expected portion of avalanches per batch of $0.6$, a learning rate of $1e^{-4}$ and a batch size of $128$. The model was trained for
255 120 epochs, i.e. iterations through the entire dataset, with early stopping when the class-separation metrics started decreasing. Additionally, we applied data augmentation by randomly shifting the $10\,\text{s}$ window signals by 0 to 1 seconds to either the right or left, to reduce overfitting in the avalanche class and for better generalization (Zhu et al., 2020). Similarly, in the spectral autoencoder, we used an expected portion of $0.5$ avalanches, a learning rate of $1e^{-4}$ and a batch size of $128$ and found 5 training epochs to be optimal.





## 4.3 Feature classification

The motivation for separating the feature extraction and classification processes was manifold. First, the partial uncertainty in the labels led to the conclusion that an unsupervised feature extraction approach is more robust to label noise and therefore preferable, as it could additionally leverage more unlabelled data. In contrast, a fully supervised neural network might suffer from the relatively low number of labels and bias, tending to overfit these expert labels rather than learn avalanche characteristic patterns in seismic signals. In an early stage, we tested this approach and did not observe better results. Thus for better comparability of the features themselves with the benchmark model, i.e. feature engineering, we pursued the unsupervised feature extraction approach. Ideally, several classifiers can then be used, combined or ensembled over different feature extraction steps.

Apart from expert labels, we considered the subjectivity of the manual cuts, the attenuation of avalanche signals with the distance to the sensors and the low initial energy of avalanches, with which we inevitably included 10 s windows from avalanche signals, which rather account towards background noise. This particularly applies to the starting and ending sections of a signal (see the upper plot in Fig. 2). Labelling these parts as avalanches (false positives) bears the danger of distracting a fully supervised neural network. Therefore, we decoupled the classification from the unsupervised feature extraction and implemented random forest classifiers for each feature set. The random forest model is a widely used algorithm for classification in general and for seismic event detection (e.g. Li et al., 2018; Provost et al., 2017; Chmiel et al., 2021), as it is favourable when dealing with high-dimensional features and heterogeneous (seismic attributes) input data and it provides output probabilities estimates.

The random forest algorithm was introduced by Breiman (2001) and belongs to the class of ensemble methods. During training, several decision trees (estimators) are grown. Each tree is grown on a different bootstrap sample of the original dataset, i.e. a random draw with replacement. Instead of using the entire set of features (columns) in the original dataset, a random subset is assigned to each node in the tree individually. The split (branch) is based on a single feature from this random subset, which is optimal under a specified splitting criterion, such as the Gini information criterion when dealing with categorical (classification) splitting problems. During inference, each tree prediction is aggregated to form a final majority vote, from which it is possible to retrieve class proportions, often interpreted as probabilities.

In search of the best hyper-parameters of this tree-growing algorithm, e.g. the maximal number of estimators (trees), we used a randomized grid search with 3-fold cross-validation. This method evaluates hyper-parameter combinations by fitting the random forest model to two of the three train folds and testing it on the left-out fold. As a scoring function, we chose the avalanche class f1-score to weigh the precision and recall uniformly. Finally, we averaged the performance across the three folds. This optimization process was applied with the three feature sets individually, i.e. the seismic attributes and the autoencoder features, to find the random forests presented in Table E1.

## 5 Results

After completion of the model development, we evaluated the three approaches on the test fold (fold 4 in Fig. 3). First, we summarized the results of the seismic attribute, TAE and SAE feature classification on the windowed 10 s seismic signals (Sect.



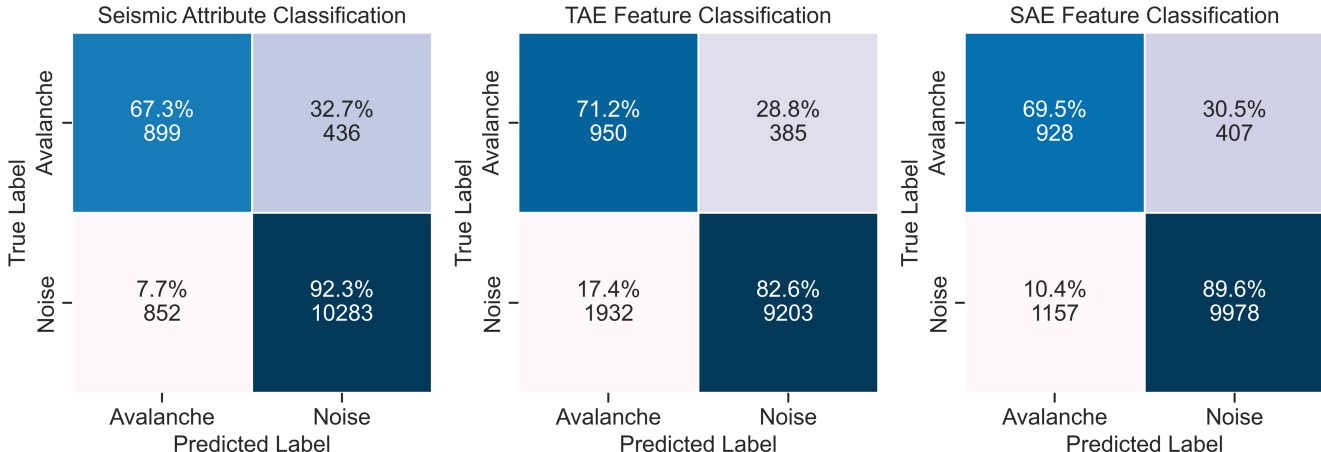

**Figure 7.** Confusion matrices of the binary classification results for the three feature sets on the held-out test fold data. The rows indicate the true (expert) labels, while the columns provide the predicted labels of the random forest classifiers. The colours code the percentage numbers.

5.1). Further, we aggregated the predictions by averaging the per-sensor 10 s window probabilities over the seismic array (Sect. 5.2). Thus, we gained insights into the predictions of unique 10 s signals at our study site.

## 5.1 Single sensor predictions

The true positive rates (or avalanche recall) were similar across the models (Fig. 7), i.e. between 67.3% and 71.2%. Neverthe-less, the avalanche recall was slightly higher for the autoencoder features classification. Regarding the true negative rates (or specificities), i.e. the probability that an actual noise event will be predicted as noise, we noted that the TAE features classi-fication showed the lowest rate of 82.6% and also showed the lowest avalanche precision of 0.33, compared to 0.51 for the seismic attributes and 0.45 for the spectral autoencoder features (Table 1). Thus, we expect this model to produce comparably more false alarms (false positives). Overall, the macro-average f1-score reached values of 0.76, 0.67 and 0.74 for the seismic attributes, the TAE features and the SAE features classification respectively (Table 1). Additionally, since feature extraction and its information content are core concepts of this study, we visualized the part of the latent spaces in Fig. 8. As earthquakes are a significant proportion of the noise class and labels were available, we show them separately. This visualization provided some insights into the organization of the autoencoder latent space.

## 5.2 Array-based predictions

In addition to the predictions on the individual 10 s windows, we aggregated the window predictions over the 5 sensors in the seismic array by averaging the per-sensor output probabilities, resulting in improved model performance (Fig. 9). The macro-average f1-score increased by 2.6% (seismic attributes), 4.5% (TAE) and 5.4% (SAE). After ensembling, the seismic attribute and the SAE feature classification yielded similar performance in the classification metrics (see Table 2). Despite this





**Table 1.** Classification metrics on the (unseen) test fold data for the three feature sets. Due to the strong class imbalance, the weighted averages of the metrics are not shown.

| Model | Class | Precision | Recall | F1 | Support |
|---|---|---|---|---|---|
| Seismic Attribute | Avalanche | 0.51 | 0.67 | 0.58 | 1335 |
| | Noise | 0.96 | 0.92 | 0.94 | 11135 |
| | Macro Avg | 0.74 | 0.80 | 0.76 | 12470 |
| | Accuracy | | | 0.90 | |
| TAE Features | Avalanche | 0.33 | 0.71 | 0.45 | 1335 |
| | Noise | 0.96 | 0.83 | 0.89 | 11135 |
| | Macro Avg | 0.64 | 0.77 | 0.67 | 12470 |
| | Accuracy | | | 0.81 | |
| SAE Features | Avalanche | 0.45 | 0.70 | 0.54 | 1335 |
| | Noise | 0.96 | 0.90 | 0.93 | 11135 |
| | Macro Avg | 0.70 | 0.80 | 0.74 | 12470 |
| | Accuracy | | | 0.87 | |

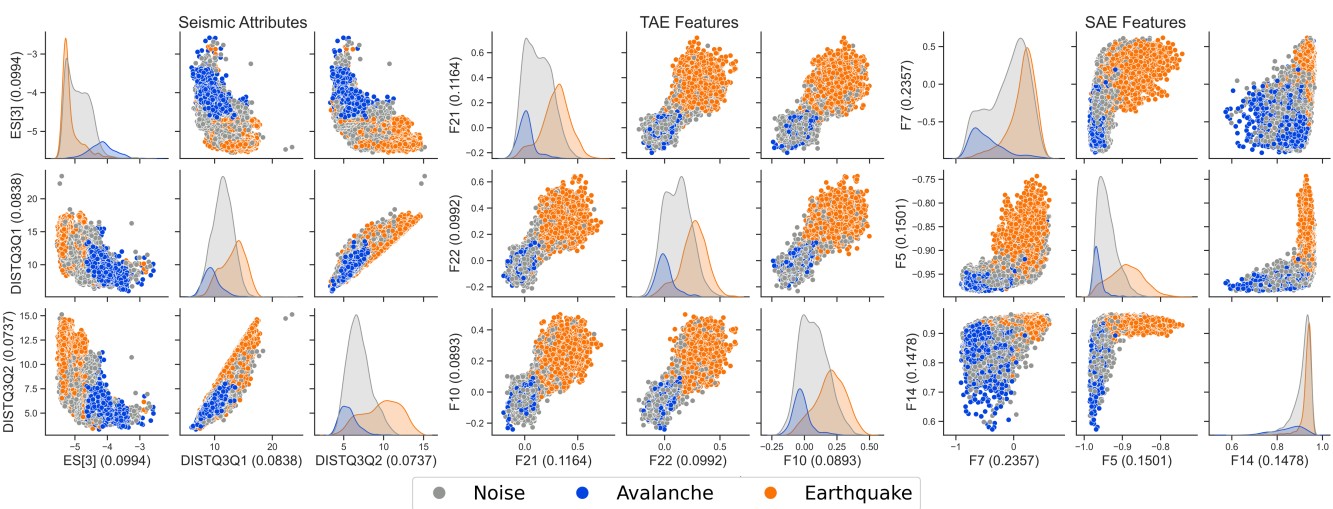

**Figure 8.** Latent space visualization of the most important features according to the impurity-based feature importance of random forest models for the seismic attributes (left), the temporal autoencoder features (middle) and the spectral autoencoder features (right). In parenthesis, the impurity-based importance of each feature is shown.

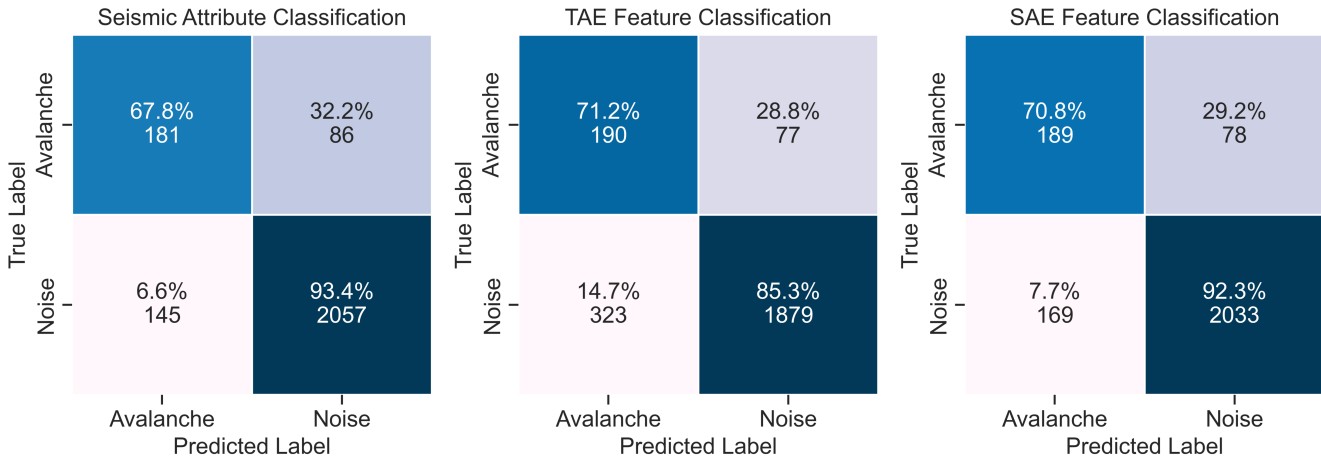

**Figure 9.** Results on the held-out test set after applying a probabilistic aggregation of the 10 s predictions over the 5 sensors of the array. The rows indicate the true (expert) labels, while the columns provide the predicted labels of the random forest classifiers. The colours code the percentage numbers.

improvement, the TAE feature classification still showed approximately double the number of false alarms, i.e. 323 (14.7%), compared to the other models. The array-based aggregation further enabled us to investigate how predictions over an entire seismic signal evolve across the array (Fig. 10). For the avalanche shown in Fig. 1 and 2, the models are comparably unsure in the starting phase, i.e. when it emerges from background noise. However, as the signal becomes more energetic, the avalanche probability increases for all models.

### 5.3 Event-based predictions

Besides the single sensor and array-based predictions (Sect. 5.1 and 5.2), we investigated the predictions on the event level to close the gap to avalanche activity assessment and provide a broader outlook. This for, we aggregated the array-level predictions in Fig. 9 over the entire duration of an event. We applied the rule that if at least two consecutive windows (or 15 s of an event) were positively predicted, the entire event was positive, i.e. an avalanche. This threshold of two windows was not optimized. However, considering that the shortest avalanche in the dataset is 13 s, this boundary was feasible. This post-processing led to the results in Appendix 5.3. Figure F1 shows a significant increase in avalanche recall with values of 81.8% (seismic attributes), 87.9% (TAE) and 5.4% (SAE). Nevertheless, the overall performance of the three models decreases by about 5% (see Table F4).

### 6 Discussion

So far, we compared the performance of a human-engineered seismic attribute classification approach and the autoencoder feature classification results based on a dataset containing 10 s seismic signals on a single sensor-level and multiple sensor-



**Table 2.** Classification metrics on the test fold data set after probabilistic aggregation over the 5 sensors. Due to the strong class imbalance and bias towards the noise class, the weighted averages of the metrics are not shown.

| Model | Class | Precision | Recall | F1 | Support |
|---|---|---|---|---|---|
| Seismic Attributes | Avalanche | 0.56 | 0.68 | 0.61 | 267 |
| | Noise | 0.96 | 0.93 | 0.95 | 2202 |
| | Macro Avg | 0.76 | 0.81 | 0.78 | 2469 |
| | Accuracy | | | 0.91 | |
| TAE Features | Avalanche | 0.37 | 0.71 | 0.49 | 267 |
| | Noise | 0.96 | 0.85 | 0.90 | 2202 |
| | Macro Avg | 0.67 | 0.78 | 0.70 | 2469 |
| | Accuracy | | | 0.84 | |
| SAE Features | Avalanche | 0.53 | 0.71 | 0.60 | 267 |
| | Noise | 0.96 | 0.92 | 0.94 | 2202 |
| | Macro Avg | 0.75 | 0.82 | 0.77 | 2469 |
| | Accuracy | | | 0.90 | |

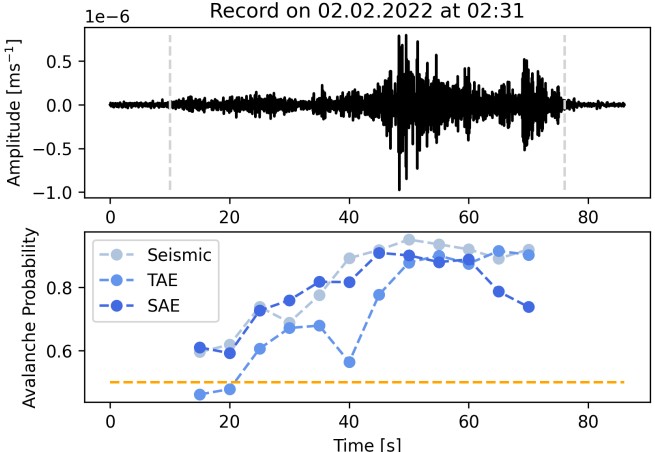

**Figure 10.** Example of the seismic signal generated by an avalanche (up) and the mean output probabilities for each developed model over the entire avalanche signal (down). The probability is computed as the average of the individual probabilities predicted by each sensor every 5 seconds (10 s windows with 50% of overlap). The manual cuts are highlighted in dashed grey lines (upper plot), and the classification threshold 0.5 is in orange (lower plot).



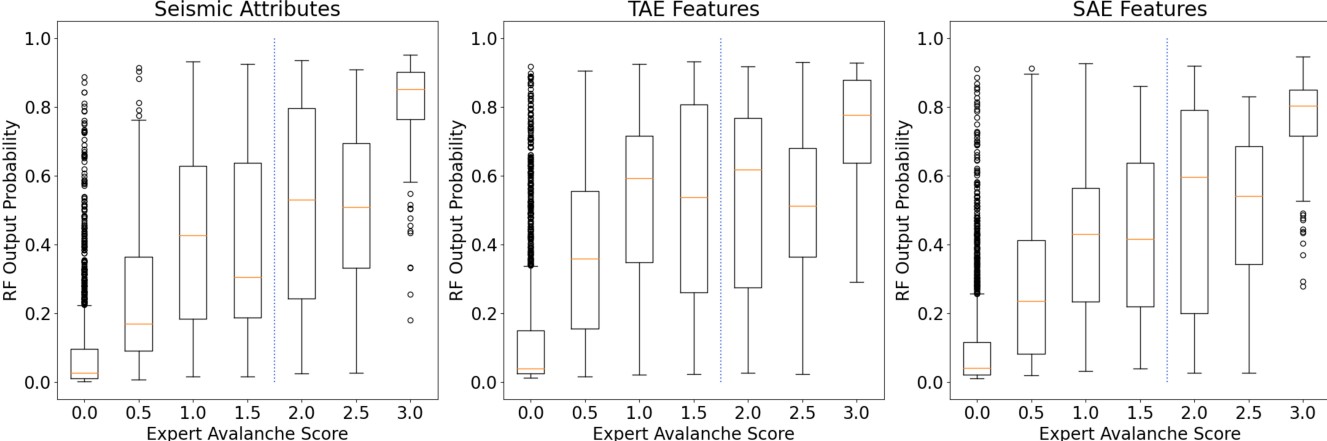

**Figure 11.** Array-based output probabilities of the random forest models for their respective input features with expert avalanche scores. The blue dashed line indicates the threshold applied to the expert scores to assign avalanche class labels.

level (aggregation). With the latter aggregation, we observed a significant reduction in false alarms and a slight improvement in recall for the avalanche class. Furthermore, we noticed that the automatically learned features, and specifically the ones from

the spectral autoencoder, performed better than the seismic attributes. Hence, the results showed that spectral input information seemed favourable. In the following, we contextualise the results by investigating the detection errors and their possible origins. Therefore, we summarize the model development (Sect. 6.1) and dived into the false predictions of the models to find potential limitations and reasons (Sect. 6.2 and 6.3). Finally, we compared the results to previous works (Sect. 6.4).

## 6.1 Model

Machine-learning models are strongly influenced by the quality and size of the dataset. The relatively small size constrained us to design autoencoder architectures with rather few trainable weights. In addition, we used each sensor independently to compensate for dataset size, as each sensor can be considered as a different view of the events. However, this came at the cost of introducing correlation among dataset samples as the sensors were installed nearby (Fig. 1) and thus recorded very similar signals, yet not necessarily adding much new and enriching information to the dataset. Given that the dataset will increase in

the next years, we will consider incorporating the 5 sensors as distinct channels in a convolutional model in future studies. With this, the sensor aggregation and fusion would be implicitly implemented into the model. Another aspect to bear in mind was the input normalization. Normalizing input data has proven crucial when training neural networks (Sola and Sevilla, 1997). The temporal autoencoder, in particular, therefore loses information on absolute and relative amplitudes. Yet, both autoencoders could still capture signal characteristics and remarkably show similar patterns when looking at continuous predictions (see Fig.

10). Alternatively, a normalization over the entire signal before windowing could be envisioned to preserve information on relative amplitudes. However, this is not practical for (near) real-time signal processing.





Further, the dataset drove the decision to separate the feature extraction and classification. The unsupervised feature extraction is not constrained to a labelled dataset (only the model selection and hyperparameter tuning are), an advantage when dealing with non-ground-truth labels (two-thirds of the avalanches were not verified). The performance of the classifier is then

decoupled from the feature extraction. This allowed us to analyze a lower-dimensional embedding of the dataset by inspecting the feature space distributions (Fig. 8). Here, we visualised the earthquake class separately, as earthquake and avalanche signals can be similar in the time domain (Heck et al., 2019a), which we, thus, wanted to investigate in the feature domain. We also had labels for earthquakes simplifying the visualization. In an early stage, we trained models with three classes (earthquake separately), without seeing an increase in overall model performance. Moreover, note that training a model to also classify

earthquakes was out of scope as these can be detected with other methods. Overall, the three event types, i.e. avalanches, earthquakes and rest, varied in the encoding locations, yet also showed considerable overlap. Interestingly though, the avalanche and earthquake signals were well separated (blue and orange in Fig. 8). The rest (grey) resembled a connecting cloud between avalanche and earthquake signals. The reason for this might be two-fold; first, the heterogeneity of these noise events by potentially comprising minor avalanches and low magnitude earthquakes (false negatives), and second, the strong attenua-

tion in some sections of avalanche signals resulting in low amplitude avalanche windows. The heterogeneity within the noise class originated from including different sources in comparable amplitude ranges, e.g. earthquakes, aeroplanes or strong wind. However, the different types of seismic sources of comparable amplitude range are definitive to be expected and need to be considered in a real-time detection system.

Finally, the applied expert labelling was subject to an unknown degree of subjectivity and belief for the non-verified events.

In addition, having decided upon a hard threshold to convert expert scores to class labels further blurred the boundaries between the avalanche and noise class, i.e. the noise class might include minor avalanches (false negatives). We, therefore, investigated the relationship between the random forest's output probabilities and the expert scores of potential avalanche signals (Fig. 11). Also, we found the average expert agreement rate on the avalanche samples to be 58%, i.e. on average, two experts agree on 58% of the avalanche signals. Overall, the output probabilities of the random forest models positively increased

with the expert scores. As expected, we also noted the highest uncertainty at the selected threshold (dotted blue line). When comparing the feature sets, the classification with the seismic attributes yielded clearer steps over expert scores and more distinctive probabilities for the highest and lowest expert scores. A measure to mitigate having to deal with noisy labels in future works might be to solely include verified avalanches and discard the non-verified ones for training the autoencoders. Another noticeable observation, which bridges to the upcoming Sect. 6.2 and 6.3, was the number of outliers for the expert

scores of 0.5 (false positives) and 3.0 (false negatives), most prominently in the seismic attributes classification.

## 6.2 Missed avalanche windows

Two types of errors are inherent in a binary classification problem, namely false negatives (FNs) and false positives (FPs), which are the focus of this and the following Sect. 6.3 respectively.

Looking again at Fig. 11, we accredited the outliers in the expert score of 3.0, i.e. FNs, to the nature of avalanche signals.

Concretely, avalanche signals slowly emerge from the background noise due to source-receiver distance and the low generation





of energy in the initial and very end stages of avalanche motion, resulting in the typical spindle-shape signal with a relatively low signal-to-noise ratio at the beginning and end of the signal (Suriñach et al., 2001; van Herwijnen and Schweizer, 2011; Pérez-Guillén et al., 2016). We suspect that the models had difficulties correctly classifying these parts of an avalanche signal, producing FN predictions. Further, the manual cutting was rather generous in including the entire avalanche signal with parts

characterised by very low amplitudes. The selection of the initial and end of the signals was subjective, and we cannot exclude that some background noise was included. For instance, Fig. 12 a) shows a comparison of the time series of array-based averaged predictions for each model with the misclassified onset of an avalanche event, while in Fig. 12 b), the end portion was characterised by a very low signal-to-noise ratio. In Fig. 12 a), the first few time windows from 10 s to 35 s are arguably rather noise, as suggested by the model probabilities. Tough as the signal strength increases, model probabilities also increase.

Concretely, if we considered the first five predictions or time windows as noise, this sample accounts for 5 (non) FNs in the results in Fig. 9 and approximately 25 in Fig. 7 per model. The array-based prediction aggregation did not reduce these missed 'avalanche' windows (Fig. 9) since all the sensors predicted low probabilities of being an avalanche. Thus, we were left with approximately one-third of FNs in all three models. In a potential early-warning operation, an effective model should be able to detect all signal parts generated by avalanches, particularly the onset, to identify the avalanche movement in its early stages

and trigger a corresponding alarm. Thus, as the models tend to miss the start of an avalanche, the current classifiers might not be suited for avalanche warning. In addition, when trying to assess the overall avalanche activity, missed avalanches are not favourable. Installing an additional sensor near the release area and avalanche path could address this issue. However, considering the characteristics of our test site (Fig. 1), where avalanches can flow over multiple paths, a single sensor will not be enough for the detection of all the avalanches.

For a general outlook, we further post-processed the array-based predictions (Fig. 9) to formulate event-based predictions. We considered an entire signal an avalanche if at least two consecutive windows (i.e. 15 s that is approximately the minimum duration of an avalanche signal) were positively predicted. In theory, this should eliminate the FNs in the tails of the actual signal and provide us with event-based detectors. For instance, in Fig. 12, we then would detect avalanches with this post-processing. Indeed, in Fig. F1, we observe a drastic reduction in missed avalanches for the three models, which achieved a high

avalanche recall of 0.82 (seismic attributes), 0.88 (TAE) and 0.91% (SAE).

In closing, we reduced the missed avalanches by applying the presented post-processing steps. Furthermore, we observed that the models struggle to detect the starting and ending of an event (Fig. 12). We argued that this behaviour is reasonable and in part desirable as these parts of an event often resemble background noise. However, in most cases, the entire (unique) event is detected (Fig. F1). Thus, the models could, in turn, be considered to annotate large datasets, which in turn can be used to

detect fine precursor signals.

### 6.3 False alarms

The second type of error, i.e. false positives (FPs) or false avalanche alarms, showed greater variation in numbers across the three models. With 7.8% the seismic attributes produced the smallest portion of false positives. Predicting with the TAE features resulted in roughly three times as many false positives, with the SAE feature prediction in between. However, we observed a

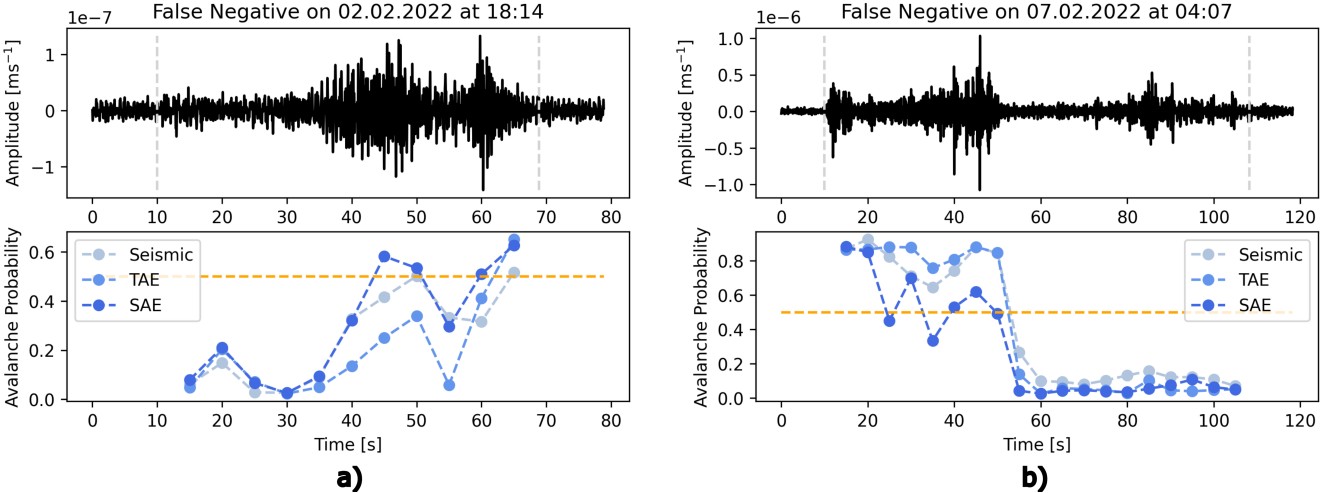

**Figure 12.** Signals generated by avalanches triggered on 2 February 2022 at 18:14 (top left) and 7 February 2022 at 04:07 (top right) and comparison of the array-based averaged probabilities by each model over the entire length of the avalanche signals (bottom). The dashed vertical lines in grey indicate the manual cuts.

more significant improvement in these errors when aggregating over the array (Fig. 9). This suggested that the 5 recordings of a specific event, particularly noise events, can show strong variations across the array, which we filtered by this averaging. As the noise class is extremely dominant and, for instance, 10% FPs result in approximately 1000 FP samples (compared to 1335 avalanche samples), the avalanche precision of all three models is relatively low with 0.51 (Seismic Attributes), 0.33 (TAE) and 0.45 (SAE).

We therefore analyzed the origins of FPs to find potential tendencies or failure cases (Fig. 13). Most FPs, i.e. 76% (seismic attributes), 65% (TAE) and 71% (SAE), were generated by windows either carrying a non-zero avalanche score or belonging to an earthquake. Interestingly, the highest portion of false positives falls to windows with an avalanche score of 0.5, i.e. 'one' expert thinks it might be an avalanche. This might indicate that minor-size avalanches, or larger avalanches that flowed at the detection limits of the system, are not well recognized by the experts yet by the models. Considering the earthquakes, the test 425 fold comprises a total of 3880 earthquake windows, of which only 135 (Seismic), 200 (TAE) and 158 (SAE) are misclassified as avalanches, i.e. 3.5%, 5.2%, 4.1%. The remaining approx. 30% FPs in all models originated from unknown sources.

    Overall, our results thus showed that using an array of sensors helped to reduce the number of false avalanche detections by averaging the predictions of the sensors. This can be viewed as model ensembling and is generally known to improve results (Mohammed and Kora, 2023). Second, including features from the frequency domain tended to show fewer FPs. Third, 430 an interesting and positive finding was that the models rarely confused earthquakes for avalanches (on average 4.3% of all earthquake windows). Finally, the models generate false alerts to a similar extent to previous studies in avalanche detection (e.g. Bessason et al., 2007; Rubin et al., 2012; Hammer et al., 2017; Heck et al., 2018). Thus, they might not yet be suited for an early-warning application. However, the models could be implemented in an avalanche activity assessment process or to label



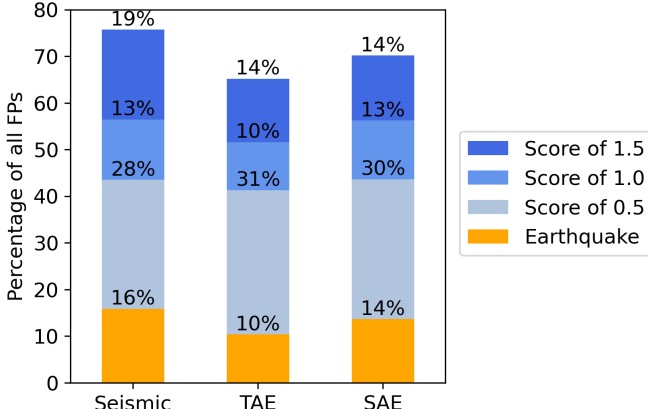

**Figure 13.** Analysis of origins for false positives as a percentage of the total amount of false positives per model.

unverified events in the future by being aware of the limitations and that they tend to produce too many avalanche detections.
In pursuit of reducing the number of false alerts, one might consider including other types of recordings, e.g. infrasound data (Mayer et al., 2020). Also, implementing specialized data augmentation techniques to increase the variety and number of the avalanche recordings, e.g. seismic data augmentation techniques (Zhu et al., 2020) or generative models (Wang et al., 2021), might help to make the classifiers more robust.

### 6.4 Comparison to previous studies

To conclude, we put our results in a broader context by comparing them with previous studies. Provost et al. (2017) used a random forest model based on the 71 engineered seismic attributes. They reported stunning true positive rates of 94%, 93% and 94% for the rockfall, quake and earthquake class and a true negative rate of 92% for the noise class. The setting, however, is difficult to compare, as they used non-windowed signals from an evenly distributed dataset comprising 418 rockfalls, 239 quakes, 407 earthquakes, and 395 noise events. Also, these event types typically generate signals with a higher signal-to-noise
ratio than avalanches. Moreover, they included polarity and network attributes in the features, which for the classification turned out to be most important. Nevertheless, with 92% true negatives, their model is comparably prone to producing false alerts as the models in this study are. Also, for avalanche detection, several studies presented the approach of engineering features and subsequent classification (e.g. Bessason et al., 2007; Rubin et al., 2012; Hammer et al., 2017; Heck et al., 2018). Rubin et al. (2012) used 10 engineered features in the frequency domain and tested 12 classification models, of which the decision stump
classifier showed the highest overall accuracy of 93%. However, the model showed a poor precision of 13.2%, hence, producing many more false alerts. In contrast to our approach, they only considered avalanches verified on camera images or manually picked events. Heck et al. (2018) used the same avalanche catalogue of 283 avalanches, of which 25 were confirmed and the rest were labelled by three experts. They implemented engineered temporal and spectral features and used an HMM as a classifier. Similar to most previous studies, they also noted high values of FPs. Moreover, they observed improvements when aggregating



single sensor to array-based predictions as we did in this study. In conclusion, based on the results of this and previous studies, we expect that an avalanche predictor based on solely seismic data will always produce false alarms, as it remains a difficult task to identify low-energy avalanche signals. Therefore, installing a secondary seismic detection system in the proximity of the avalanche path would be advantageous in mitigating false alarms. Alternatively, integrating a complementary detection system, such as an infrasound system, could also be beneficial but less cost-effective.

In summary, the classification results met the performance of previous studies on avalanche detection. However, the core contribution of this study is two alternatives to extract features from seismic signals. We showed that the proposed encoder features are applicable for avalanche detection and compare well to engineered features. In particular, the learned feature extraction does not depend on prior expertise or knowledge and thus can be adapted easily to new settings, e.g. changing environments, without having to set some parametrisations of expert features. Moreover, with growing dataset size or larger

datasets, it can improve over time. Finally, a future interesting comparison would be to evaluate the models on how they generalize to other test sites and settings.

## 7 Conclusions

We proposed two unsupervised seismic feature extraction methods based on deep learning algorithms and a standard seismic attributes set to train three random forest classifiers for avalanche detection. The dataset was compiled from seismic avalanche

data recorded during two winter seasons in Davos, Switzerland. While in earlier studies, seismic data classification mostly followed the approach of extracting well-defined signal attributes to train classifiers, the proposed deep learning models bridge the gap to a purely learned (automatic) pipeline.

Overall, the classifiers achieved macro-average f1-scores ranging from 0.70 to 0.78 with avalanche recall values ranging from 0.68 to 0.71. Our results clearly show that including features from the frequency domain improves model performance.

As the onset and end of avalanche signals were often misclassified as noise, due to low signal-to-noise ratios, we proposed a simple post-processing step to reduce the missed avalanches by imposing that at least two consecutive prediction windows, i.e. 15 s, are positive. This criterion significantly improves the avalanche recall, ranging from 0.82 to 0.91. Lastly, contrary to previous expectations, earthquakes are rarely mistaken for avalanches at our study site.

Revisiting our primary goal of comparing human-engineered with automatic feature extraction, there is no denying that

the standard seismic attributes classification is a robust approach. These predefined attributes have been studied and applied for a decade and optimized and tuned throughout various studies. The unsupervised representation learning, in contrast, is a completely new approach to seismic avalanche data analysis. We have shown that it bears potential for future implementations and applications. Compared to engineered features, the learned features require no prior expertise and, therefore, can easily be adapted to changing environments without having to set some parametrisations of expert features. Also, they can improve with

growing dataset size in future.





*Code and data availability.* The code and data to develop the final models used in this study will be made available on GitLab and EnviDat.



**Table A1.** Detailed view on the applied splits of the dataset. For each fold, the table shows the number of respective events.

| Fold | Date | Avalanches | Earthquakes | Noise |
|---|---|---|---|---|
| 1 | 13.01.2021 - 28.01.2021 | 17 | 39 | 196 |
| 2 | 29.01.2021 - 24.05.2021 | 16 | 39 | 100 |
| 3 | 10.01.2022 - 04.02.2022 (excl. 02.02.2022) | 18 | 39 | 138 |
| 4 | 06.02.2022 - 17.05.2022 (incl. 02.02.2022) | 33 | 66 | 211 |

## Appendix A: Dataset

Table A1 depicts the date ranges in each fold and the respective number of events. We used folds 1, 2 and 3 for the cross-validation, i.e. the model development, and the test fold (number 4) to obtain the final results on unseen data. In general, we
picked the folds consecutive in time, with a minor exception in the test fold, where we moved the 2nd of February from fold 3 to the test fold. This balanced the number of events in the folds more evenly.

## Appendix B: Seismic attributes

The implemented engineered feature extraction follows the work of Provost et al. (2017) and Turner et al. (2021). In contrast to these, by defining our frequency band to 1-10 Hz we modified the attributes correspondingly. Also, we discarded network or
polarity-related attributes as we developed individual models per sensor, and most of our sensors only contained one vertical component. In summary, we extracted 22 waveform attributes (Table B1), 17 spectral (Table B2) and 18 spectrogram attributes (Table B3).

## Appendix C: Metrics

Besides the classifiers, we also evaluated the unsupervised clustering of the autoencoders in the latent space. Therefore, we
used the classification and clustering metrics defined here.





**Table B1.** Waveform attributes extracted from the 10 s seismic signals.

| Number | Description |
|--------|-------------|
| $1-2$ | Ratio of the mean and median over the maximum of the normalised envelop signal |
| 3 | Ratio between ascending and descending time |
| 4 | Kurtosis of the raw signal |
| 5 | Kurtosis of the envelope |
| 6 | Skewness of the raw signal |
| 7 | Skewness of the envelope |
| 8 | Number of peaks in the autocorrelation function |
| 9 | Energy in the first third part of the autocorrelation function |
| 10 | Energy in the remaining part of the autocorrelation function |
| 11 | Ratio of 10 and 9 |
| $12-16$ | Energy of the signal filtered in $[1,3], [3,6], [5,7], [6,9]$ and $[8,10]$ Hz |
| $17-21$ | Kurtosis of the signal in $[1,3], [3,6], [5,7], [6,9]$ and $[8,10]$ Hz |
| 22 | RMS between the decreasing part of the signal and $I(t) = Y_{max} - \frac{Y_{max}}{t_f - t_{max}} t$ |

## C1 Classification metrics

Various metrics exist to evaluate binary classification problems and are all tailored to specific objectives. For instance, the precision is chosen when false alerts, i.e. false positives, are critical, the recall is sensitive to missed events, i.e. false negatives and the f1-score combines both to form the harmonic mean of both as follows:

$$F1 = 2 * \frac{Precision * Recall}{Precision + Recall} \tag{C1}$$

The macro average summarizes the per-class results within a single value. This value is an unweighted mean over the given classes and ensures that the values are not biased towards the most frequent class, i.e. noise.





**Table B2.** Spectral attributes extracted from the 10 s seismic signals. The Nyquist frequency (NyF) is 100 Hz, i.e. half of the sampling rate.

| Number | Description |
|---|---|
| $23-24$ | Mean and Max of the FFT |
| 25 | Frequency at the maximum |
| $26-27$ | Central frequency of the 1st quartile and 2nd quartile |
| $28-29$ | Median and Variance of the normalized FFT |
| 30 | Number of peaks |
| 31 | Number of peaks in the autocorrelation function |
| 32 | Mean value for the peaks |
| $33-36$ | Energy in $[\frac{1}{100},\frac{1}{4}]NyF$, $[\frac{1}{4},\frac{1}{2}]NyF$, $[\frac{1}{2},\frac{3}{4}]NyF$ and $[\frac{3}{4},1]NyF$ |
| 37 | Spectral centroid |
| 38 | Gyration radius |
| 39 | Spectral centroid width |

$$Macro-F1 = \frac{1}{K} * \sum_{k=0}^{K} F1_k \,, where\, K = 2 \qquad (C2)$$

**C2  Clustering metrics**

A natural metric choice when evaluating different autoencoders is a reconstruction loss, e.g. the mean squared error on which the autoencoders in this work were trained. In pursuit of good autoencoder features for later classification, however, we aimed to optimize the latent space representation. Since a good reconstruction does not necessarily imply a sufficient separation in latent space, we explored clustering metrics to compare the latent space distribution of different models with the given (expert) labels. We, therefore, implemented the silhouette score (Rousseeuw, 1987) and the Calinski–Harabasz index (Caliński and Harabasz, 1974). These scores are usually used to evaluate clustering algorithms that predict classes, e.g. k-means. The silhouette score computes the mean intra-cluster and inter-cluster distances per sample. For instance, given a sample, it calculates the distance to the cluster it is part of (a) and the distance to the nearest cluster it is not part of (b) and forms the sample score:

$$S_i = \frac{b - a}{max(a,b)} \qquad (C3)$$

After taking the mean over all samples, the silhouette score ranges from -1 (worst) to 1 (best). The Calinski–Harabasz index, or variance ratio criterion, on the other hand, is the ratio of between- and within-cluster dispersion. The between-cluster dispersion



is defined as the weighted sum of squared Euclidean distances of the cluster centroids and the overall centroid (higher, better), and the within-cluster dispersion is given as the sum of squared Euclidean distance of the samples and their respective cluster centre (lower better). Thus, a good clustering algorithm is supposed to yield a high Calinski–Harabasz score.

## Appendix D: Weighted random sampler

Training a deep learning model on a dataset characterised by a severe class imbalance can bias the model predictions towards focusing solely on the most frequent class. The model can thus achieve high accuracy by accurately predicting this class. Therefore, it can fail to predict events in the minority class, which in our study is the most interesting one, i.e., avalanches. To mitigate this problem, we applied a weighted bootstrapping technique during the training of the autoencoders, a so-called weighted random sampler, as implemented in PyTorch (Paszke et al., 2019). Therefore, we assign the following weights to 530 each sample of the avalanche ($w_{av}$) or noise class ($w_{no}$).

$$w_{av} = \frac{N_{no}}{N_{av}} \frac{P_{av}}{1 - P_{av}}; \qquad\qquad w_{no} = 1 \qquad\qquad\qquad (D1)$$

$P_{av}$ is the user-defined portion of expected avalanches within each batch. Internally, these weights are rescaled and interpreted as probabilities.

## Appendix E: Random forest optimization

The random forest models and their optimizations were implemented using the scikit-learn library (Pedregosa et al., 2011). Table E1 presents the three selected random forest models that were optimized on the same hyper-parameters grid and ranked based on the avalanche class f1-score.

## Appendix F: Autoencoder optimization

To select the autoencoder hyper-parameters, we opted to first optimize model intrinsic parameters, e.g., hidden dimensions or 540 the number of layers, instead of training strategy parameters. This separation reduced the computation time.

The temporal autoencoder architecture optimization proved to be more sensitive and critical. First, we optimized the kernel size, stride, number of filters, feature dimension and activation function. We observed that the kernel size and stride combinations of (20, 10) and (8, 4) showed the best clustering metrics. Moreover, concerning the non-linear activation, the leaky ReLU outperformed the Tanh function in most tests. Since the overall performance was not entirely satisfying, we tested the 545 weighted random sampler (Sect. D with 50% expected avalanches in each batch. This addition to the training strategy showed a considerable improvement for most models with kernel size 20 and stride 10. Although using a kernel size of 8 and stride of 4 tended to show better clustering metrics, the reconstruction of the signals was comparably poor. Based on these observations, we implemented a kernel size of 20 and stride of 10. Also, we found the feature dimension 32 better suited than 64 or 16.



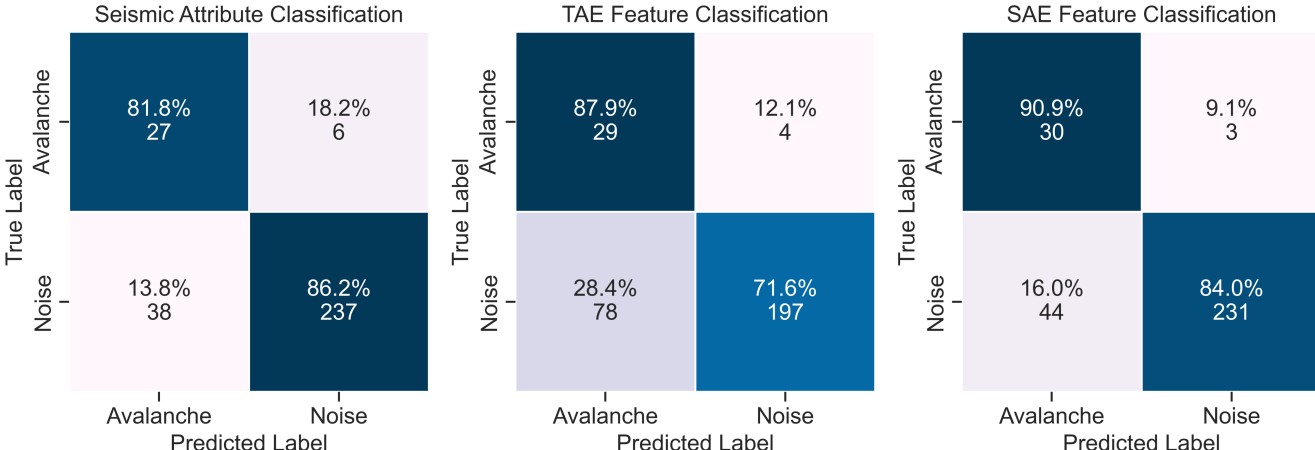

**Figure F1.** Confusion matrices of the results for the three feature sets aggregated on an event basis. The rows indicate the true (expert) labels, while the columns provide the predicted labels of the random forest classifiers. The colours code the percentage numbers.

Lastly, we selected the number of filters as 32, 64 and 128 within the encoder. See Table F2 for a summary of the best 10
models of this process and Table F1 for the selected autoencoders. Having defined the intrinsic parameters, we tested different training strategies. In particular, we optimized the learning rate, the batch size and the expected portion of avalanches in a batch. This test led to values of $1e^{-4}$, 128 and 0.6 for the temporal autoencoder. Finally, we found that augmenting the data by randomly shifting input samples by 0 to 1 s to the left or right helps.

While optimizing the spectral autoencoder, we found faster convergence. We started by testing combinations of the number
of layers with hidden dimensions, feature dimensions and activation functions. The results for the best 8 models are shown in Table F3. We foremost noted that 16 features were optimal for this task. Moreover, we observed that the Tanh activation function was favourable in comparable architectures. Finally, we selected the model highlighted in orange since it showed a good compromise between the number of weights of the network and performance. Following the same training strategy as for the temporal autoencoder, we optimized the learning rate, the batch size and the expected portion of avalanches in a batch. In
contrast to the temporal autoencoder, we used an expected portion of 0.5 avalanches within a batch, a learning rate of $1e^{-4}$ and a batch size of 128.

**F1   Event-based prediction results**

*Author contributions.*  AS: concept and design of the study, data collection and curation, model development, computational framework, analysis, writing, CP: concept and design of the study, data collection and curation, model development, computational framework, anal-
ysis, writing, reviewing, MV: concept and design of the study, model development, analysis, reviewing, CS: data collection and curation, computational framework, reviewing, AH: concept and design of the study, data collection and curation, analysis, reviewing



*Competing interests.* The authors declare that they have no conflict of interest.

*Acknowledgements.* This study was supported by a grant from the Innosuisse - Swiss Innovation Agency (37619.1 IP-ENG). We thank Prof. Fernando Perez-Cruz for the helpful discussions and numerous colleagues from SLF for help with fieldwork and maintaining the
instrumentation. We thank Geoprevent, and in particular Lino Schmid and Johannes Gassner, for sharing the radar data with us and helping with the interpretation.



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

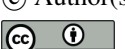



**Table B3.** Spectrogram attributes extracted from the 10 s seismic signals.

| | |
|---|---|
| 40 | Kurtosis of the maximum of all Fast Fourier Transforms (FFTs) over time |
| 41 | Kurtosis of the maximum of all FFTs as a function of time |
| 42 | Mean ratio between the maximum and the mean of all FFTs |
| 43 | Mean ratio between the maximum and the median of all FFTs |
| 44 − 46 | Number of peaks in the curve showing the temporal evolution of the FFTs maximum (44), mean (45) and median (46) |
| 47 | Ratio between 44 and 45 |
| 48 | Ratio between 44 and 46 |
| 49 | Number of peaks in the curve of the temporal evolution of the FFTs central frequency |
| 50 | Number of peaks in the curve of the temporal evolution of the FFTs maximum frequency |
| 51 | Ratio between 50 and 51 |
| 52 | Mean distance between the curves of the temporal evolution of the FFTs maximum frequency and mean frequency |
| 53 | Mean distance between the curves of the temporal evolution of the FFTs maximum frequency and median frequency |
| 54 | Mean distance between the 1st quartile and the median of all FFTs as a function of time |
| 55 | Mean distance between the 3rd quartile and the median of all FFTs as a function of time |
| 56 | Mean distance between the 3rd quartile and the 1st quartile of all FFTs as a function of time |
| 57 | Number of gaps in the signal |



**Table E1.** Selected Random Forest Models

| Parameter | Seismic Attributes | TAE | SAE |
|---|---|---|---|
| Number of Estimators | 512 | 512 | 512 |
| Maximum Depth | 8 | 8 | 8 |
| Maximum Number of Features | log2 | sqrt | sqrt |
| Maximum Number of Samples | 0.1 | 0.2 | 0.2 |
| Class Weight | Balanced | | |
| Criterion | Gini | | |
| Bootstrap | True | | |

**Table F1.** Selected Autoencoders

| Parameter | Temporal Autoencoder | Spectral Autoencoder |
|---|---|---|
| Number of Weights | 514'337 | 81'330 |
| Feature Dimension | 32 | 16 |
| Hidden Dimension | [200, 20, 2] | [139, 78, 16] |
| Filters | [32, 64, 128] | - |
| Number of Layers | 3 | 3 |
| Kernel Size | 20 | - |
| Stride | 10 | - |
| Expected Avalanche Portion in Batch | 0.6 | 0.5 |
| Learning Rate | $1e^{-4}$ | $1e^{-4}$ |
| Batch Size | 128 | 128 |





**Table F2.** The Table summarizes the TAE hyper-parameter optimization. It shows only the models for which all three metrics are ranked in the top 20. The best metrics are highlighted in bold, and the selected model architecture is in orange.

| Weights | Filters in first Layer | Feature Dimension | Kernel Size | Stride | Expected Avalanche Portion | Augmentation | Silhouette Score | Calinski–Harabasz Index | MSE |
|---|---|---|---|---|---|---|---|---|---|
| 109865 | 8 | 64 | 8 | 4 | default | False | **0.191** | **849.959** | 0.078 |
| 109865 | 8 | 64 | 8 | 4 | 0.5 | False | 0.024 | 357.494 | 0.073 |
| 109865 | 8 | 64 | 8 | 4 | 0.5 | True | 0.018 | 345.684 | 0.076 |
| 156945 | 16 | 32 | 20 | 10 | 0.5 | False | 0.033 | 374.174 | 0.06 |
| 156945 | 16 | 32 | 20 | 10 | 0.5 | True | 0.011 | 567.276 | 0.055 |
| 514337 | 32 | 32 | 20 | 10 | default | True | -0.072 | 368.876 | **0.054** |
| 514337 | 32 | 32 | 20 | 10 | 0.5 | False | 0.061 | 333.174 | 0.061 |
| 514337 | 32 | 32 | 20 | 10 | 0.5 | True | 0.041 | 613.917 | **0.054** |
| 625185 | 32 | 64 | 20 | 10 | 0.5 | False | -0.095 | 292.78 | 0.063 |
| 625185 | 32 | 64 | 20 | 10 | 0.5 | True | -0.105 | 307.477 | 0.064 |

**Table F3.** The Table summarizes the SAE hyper-parameter optimization. It shows only the models for which all three metrics are ranked in the top 10. A hidden dimension of 0.0 indicates that the dimensions in the layers of the encoder linearly decrease from the input dimension (200) to the feature dimension. The best clustering metrics are highlighted in bold, and the selected model architecture is in orange.

| Weights | Layers | Feature Dimension | Activation Function | Hidden Dimensions | Silhouette Score | Calinski–Harabasz Index | MSE |
|---|---|---|---|---|---|---|---|
| 47552 | 2 | 16 | Tanh | 0.0 | 0.227 | 1205.952 | 0.014 |
| 47552 | 2 | 16 | leaky ReLU | 0.0 | 0.218 | 1088.234 | 0.012 |
| 70880 | 2 | 64 | Tanh | 0.0 | 0.198 | 999.475 | 0.014 |
| 81330 | 3 | 16 | Tanh | 0.0 | 0.224 | **1237.579** | 0.013 |
| 81330 | 3 | 16 | leaky ReLU | 0.0 | 0.217 | 1015.357 | 0.012 |
| 112432 | 4 | 16 | Tanh | 0.0 | **0.238** | 1111.027 | 0.013 |
| 112432 | 4 | 16 | leaky ReLU | 0.0 | 0.223 | 1013.013 | 0.012 |
| 146120 | 5 | 16 | leaky ReLU | 0.0 | 0.223 | 968.953 | 0.012 |





**Table F4.** Classification metrics on the test fold data set after the aggregation over entire events of the array-based predictions. Due to the strong class imbalance and bias towards the noise class, the weighted averages of the metrics are not shown.

| Model | Class | Precision | Recall | F1 | Support |
|---|---|---|---|---|---|
| Seismic Attributes | Avalanche | 0.42 | 0.82 | 0.55 | 33 |
| | Noise | 0.98 | 0.86 | 0.92 | 275 |
| | Macro Avg | 0.70 | 0.84 | 0.73 | 308 |
| | Accuracy | | | 0.86 | |
| TAE Features | Avalanche | 0.27 | 0.88 | 0.41 | 33 |
| | Noise | 0.98 | 0.72 | 0.83 | 275 |
| | Macro Avg | 0.63 | 0.8 | 0.62 | 308 |
| | Accuracy | | | 0.73 | |
| SAE Features | Avalanche | 0.41 | 0.91 | 0.56 | 33 |
| | Noise | 0.99 | 0.84 | 0.91 | 275 |
| | Macro Avg | 0.7 | 0.87 | 0.73 | 308 |
| | Accuracy | | | 0.85 | |