# Peer review of "Autoencoder-based feature extraction for the automatic detection of snow avalanches in seismic data"

_Geoscientific Model Development, 2024_

## Author Comment (AC2)

*The authors have applied deep learning autoencoder models for the automatic and unsupervised extraction of features from seismic records. These extracted features were then used in classifiers to identify snow avalanches. This study presents a novel and relevant approach to enhance machine learning predictions, which could be useful not only for identifying snow avalanches but also for detecting other types of natural events. The overall methodology is well-defined, and the manuscript is well-written and easy to follow. I recommend the publication of this manuscript after the following issues are addressed*

*Major:*

*1) Given that the models tend to miss the onset of an avalanche, the authors should have included a scenario where only verified avalanches were used, excluding non-verified ones during the training of the autoencoders. While I am not suggesting that this must be incorporated in the revised version, as this conclusion emerged only after the study was completed, it is still worth mentioning.*

We would not expect the tendency to miss the onset of avalanches to be reduced when using only verified avalanches. All models tend to misclassify avalanche onsets as well as avalanche signals with a low signal-to-noise ratio. This suggests that the reason for missed onsets is rather found in the nature of mass movement signals and seismic recording. Avalanches are variable, moving sources of seismic energy, which attenuate significantly with distance. When an avalanche releases, the generation of seismic energy is typically low but increases as the flow moves downward due to the entrainment of mass and acceleration. The avalanche descent causes erosion processes, impacts with the terrain and the snow cover, and a final mass deposition, all of which are sources of seismic energy (Pérez-Guillén et al., 2016). Additionally, we expect an increase in seismic amplitudes over time due to a reduction in the source-receiver distance, as all avalanches approach the seismic array with time at our test site. Moreover, fully verified avalanches in our study are avalanches that were detected by the Doppler radar and/or verified with camera images. Some of them are small and thus, the signal-to-noise ratio is low. Installing a sensor in the avalanche release area would allow for recording the onset of avalanches with a higher signal-to-noise ratio, thus improving the performance of a model trained with this data. However, such a configuration would be limited to recording the onset of avalanches in the specific path where the sensor is installed, but not in all the avalanche paths of Dischma (Fig. 1), which can originate from different slope aspects and elevations.

Finally, unsupervised autoencoders are entirely independent of any class labels or information. Thus, by considering only verified avalanches, we would not reduce class ambiguity from the autoencoder's perspective but the dataset size and with it, valuable information might be lost. Nevertheless, we followed the reviewer's suggestion and retrained the spectral autoencoder and random forest model on only verified avalanches, i.e. avalanches that reached an expert score of 3. The comparison of both approaches is shown in the following figure. We indeed observe no improvement in the number of detected avalanche windows but a reduction.

Original results

Verified avalanches only

[Figure]

[Figure]

In conclusion, we are aware of this limitation and its significance for a potential early-warning system. For future studies aimed at developing an early-warning model, we would suggest examining the avalanche onsets in more detail and developing specialized models based on only these windows.
We will include parts of this reasoning and outlook in the final version of the manuscript.

*2) How were the machine learning algorithms implemented, including details such as programming languages and libraries used?*

The code is predominately written in Python using the PyTorch library for the autoencoder models, the random forest implementation of the Scikit-learn library, the Pandas library for handling the data and more standard Python libraries such as NumPy and SciPy. We will include this specification in the main text of the final manuscript under code and data availability.

*Minor:*

*Line 218: "The best model from the cross-validation procedure (Table F2) was composed of convolutions with kernel size 20 (or 0.1 s) and stride 10. " Was the MSE (Mean Squared Error) the primary metric used for classification?*

The mean squared error was used to develop, more specifically to train the autoencoders and optimize their reconstruction. Since autoencoders aim at reconstructing a given input signal, they require a reconstruction loss for training. The primary metric used to evaluate the classification was the avalanche class f1-score (line 286 in the preprint).

*Line 228: "As an activation function, we use the leaky rectified linear unit (leaky ReLU; (Xu et al., 2015))". Was the activation function unchanged during the hyperparameter optimization process?*

No, it was not. We included both the leaky ReLU and the Tanh activation function in the autoencoder optimization process.
We will clarify this in the revised manuscript.

*Line 318: Please replace "This for" by "For this"*

Yes, we will.

*References:*

Pérez-Guillén, C., Sovilla, B., Suriñach, E., Tapia, M. and Köhler, A., 2016. Deducing avalanche size and flow regimes from seismic measurements. *Cold Regions Science and Technology*, *121*, pp.25-41.

---

## Author Comment (AC3)

*The paper is dealing with snow avalanche detection using autoencoded seismic data. It uses one study side in Davos and events were picked and the performance tested. The article is well written and of interest for publication. However, the following should be adressed.*

*Discuss the effects of various avalanche types in relation to models and autoencoders. Explain how the findings can be applied to other study sites and what the specific considerations are in this context. That could be also highlighted in the comparison with the former studies.*

The effect of different avalanche types on the performance indeed is not analysed in this study, since no information on avalanche type or size was available. The primary goal of this study was to develop a model for all occurring avalanches regardless of their type or size. Therefore, we carefully selected the train and test set to include both wet as well as dry avalanches. A closer examination of different avalanche types will certainly be the subject of future studies with more information available and larger avalanche catalogues. Nevertheless, to explore the impact of different avalanche types on the detection performance, we divided the test set into mid-winter dry-snow conditions, 6[th] of Feb to 15[th] of April, and late-winter wet-snow conditions, 15[th] of Apr to 17[th] of May. In the following figure, we plotted the results in the form of confusion matrices for the spectral autoencoder feature classification in both periods. The results show that wet-snow avalanches were detected slightly better than dry-snow avalanches at the Davos test site for this test set. However, the model appears to produce more false alarms in the late winter season, which might be due to rising environmental noise, the reduction of the snow cover and its attenuation and the rise of a nearby stream.
As the treatment of different avalanche types was not in the scope of this study, we will not include this test in the final version.

6[th] of February to 15[th] of April       15[th] of April to 17[th] of May

[Figure]

Since the models have not been tested with data from other sites, their transferability to different locations remains speculative. Seismic signals generated by avalanches exhibit common patterns, as demonstrated by earlier studies conducted in various countries such as Switzerland (Suriñach et al., 2001; van Herwijnen et al., 2011), Norway (Vilajosana et al., 2007), France (Lacroix et al., 2012), and Japan (Pérez-Guillén et al., 2019). However, the generation of seismic energy depends not only on the characteristics of the source (such as avalanche size and type) but also on site-specific factors (such as seismic site effects due to topography and geological characteristics) and the source-receiver distance, which affects the geometrical and anelastic attenuation of the waves (based on the sensor configuration relative to the avalanche path and terrain characteristics). Therefore, proper validation of the model performance using input data from different locations is necessary to assess their transferability to other test sites. We expect variation in the performance arising from different configurations in the study site setup, sensor location and configuration as well as in the characteristics of the terrain and the avalanches. Nonetheless, we see the importance of developing models that can be used in different locations and will certainly consider it in future studies.

Also, we will state this in the revised manuscript.

*The conclusions and the further use should be more clear.*
*Please, avoid repetitions throughout the article.*

Thank you very much, we will consider these suggestions in the final and revised manuscript.

*References:*

Lacroix, P., Grasso, J.-R., Roulle, J., Giraud, G., Goetz, D., Morin, S., and Helmstetter, a. (2012). Monitoring of snow avalanches using a seismic array: Location, speed estimation, and relationships to meteorological variables. Journal of Geophysical Research, 117(F1):F01034.

Pérez-Guillén, C., Tsunematsu, K., Nishimura, K. and Issler, D., 2019. Seismic location and tracking of snow avalanches and slush flows on Mt. Fuji, Japan. *Earth Surface Dynamics*, *7*(4), pp.989-1007.

Suriñach, E., Furdada, G., Sabot, F., Biescas, B., and Vilaplana, J. (2001). On the characterization of seismic signals generated by snow avalanches for monitoring purposes. Annals of Glaciology, 32(1):268–274.

van Herwijnen, A. and Schweizer, J. (2011). Monitoring avalanche activity using a seismic sensor. Cold Regions Science and Technology, 69(2-3):165–176.

Vilajosana, I., Suriñach, E., Khazaradze, G., and Gauer, P. (2007). Snow avalanche energy estimation from seismic signal analysis. Cold Regions Science and Technology, 50(1-3):72–85.

---

## Referee Report (RR2)

**Review of "Autoencoder-based feature extraction for the automatic detection of snow avalanches in seismic data"**

**January 13, 2025**

**General comments**

The manuscript by Simeon et al. presents a novel method to automatically detect snow avalanches in seismic data collected at a test site above Davos, Switzerland. Specifically, the performance of three different algorithms is assessed: seismic attributes, temporal autoencoder, and spectral autoencoder. Based on this, they find that the inclusion of features from the frequency domain improves model performance and unsupervised autoencoders show potential as an alternative to the standard expert-based seismic attributes classification.

I believe the science behind this study is sound and aligns with the focus of Geoscientific Model Development. However, I suggest some restructuring and clarification prior to publication. In particular, the Discussion section is quite long and, as pointed out by another referee, the manuscript still contains some repetitive information. Considering Provost et al. (2017) found higher true positive rates for non-windowed signals, I believe determining the effect of different window lengths would add value to the manuscript. At least the potential effects of choosing a different window length should be discussed. I recommend the authors also take my specific comments listed below into account.

**Specific comments**

L41 – radius of several kilometres: This becomes clearer later on in the manuscript, but I would suggest being precise from the very beginning.

L46 – other types of mass movements: Consider adding a few examples.

L47 – other seismic sources "such as earthquakes"?

L71: You describe what an encoder is, but not a decoder.

L80: Why did you use 10 s windows and not, e.g., 5 s, 20 s, or maximum length of picked avalanche events?

L91: What are the advantages of a star-like pattern compared to others?

L127 – non-background noise signals: having 912 non-background noise signals that then split into avalanche and noise events is a confusing terminology. Consider using a different term for non-background noise signals.

 $L139 - Fig. \ 2b$ : The labels of the panels are missing in Fig. 2.

L148 – scores exceeded 1.5: Consider changing to was at least 2.0. Readers just quickly skimming the manuscript might misinterpret it as  $\geq 1.5$

L167 – even: change to roughly even.

L167 to 169: You differentiate between dry and wet avalanches here but never address it in the results.

L173: Sect. 4.3 referenced before 4.2.

Fig 3: Could add brackets to the y-label clearly indicating which are training and test folds.

Fig 4: Only the encoder is shown. Is the decoder part of the autoencoder not used?

L194: Why is the decoder discarded?

L264: Could include these results as supplementary material.

L280 – Gini information criterion: Missing citation.

Fig. 7: Consider adding a colour bar.

Sect. 5.1: I recommend integrating Sect. C1 into the main text. Otherwise, it is difficult to interpret the results.

Fig. 8: It is not clear what the x- and y-labels are.

Fig. 10 - up/down: Change to top/bottom.

L331 and 385: Sect. 6.4 referenced before 6.3. I recommend to first discuss the missed avalanche windows and false alarms, and then the applicability to early warning systems.

L367: Same as the first sentence of the paragraph.

L378: dotted blue line in Fig. 11.

L409 to 411: It would be interesting to see how the issue of identifying avalanche onset evolves under different signal window lengths (e.g., trading slightly later detection for overall better performance).

L441: What could these unknown sources be?

L445: Compared to previous studies, why can the models discussed here better differentiate between avalanches and earthquakes?

L466: How do the approaches of Bessason et al. (2007) and Hammer et al. (2017) compare the this study?

L504: I recommend ending with your key takeaway.

I hope the authors find my comments helpful.

Sincerely, Kevin Hank

---

## Referee Report (RR3)

**Major revisions are emboldened.**

Minor revisions are not emboldened.

**1. General**

- 1. Research questions and contributions should be described more explicitly.
- 2. Add task description (and why those tasks address your research questions)
- 3. **Improve the narrative/storyline** of your paper. Sometimes, explanations or context is missing, and "why" you have made certain decisions is often unclear. You also have very strong arguments at hand from time to time that you are not mentioning or considering.

**2. Structuring**

- 1. Sections sometimes need to be structured better.
- 2. **Context content conclusion:** For most parts, you adhere to that, but several subsections do not follow that structure. Using CCC consistently would significantly improve the flow and accessibility of your manuscript

**3. Language**

- Consistency: The paper's language is sometimes inconsistent. You often switch
  between descriptors (sensor array vs multiple-sensors). You do not refer to your
  different models consistently with the same name. For you, the synonyms are
  clear, but for the reader, this can be very confusing: It is unclear if two terms are
  referring to the same concept or (slightly) different concepts and whether that
  difference is relevant.
- 2. Explicit is better than implicit (explain things as explicitly as possible).
- 3. Consider running the manuscript through a language correction tool I am not a native speaker, but some formulations sound "off" to me.

**4. Formatting**

- 1. Make sure to add links for figure and section labels.
- 2. Make sure all figures are highly resolved.
- 3. Change tables to booktabs.
- 4. Fix appendix issues.

**5. Data**

- 1. Label incompleteness: What if experts have missed avalanches (unknown avalanche activity?)
- 2. Label uncertainty: Should the task be considered a regression task instead of a binary classification task?
- 3. Label noise: The agreement score of expert labels is relatively low. Evaluating the performance of the DL model is difficult with that.
- 4. Question: How would you evaluate the success rate of expert labeling? Can you evaluate the ML model independently of expert labels by any chance? Is there a theory on what avalanche signals look like in the Fourier domain, and could you calculate some score based on that?

**6. Data processing**

1. How do you get the data provided in your repository? I assume this is not the raw data from the sensors? Making that bit accessible / describing it would be important to improve reproducibility.

- 2. Extending the dataset: Have you considered extending your dataset with samples from previous publications? (Provost et al. (2017) + Rubin et al. (2012))
- 3. **Data split**: you should also separate locations / sensors, so you do not have leakage via correlated samples. Are you doing a stratified split? I.e., are the same amount of pos/neg samples available in each split? Or is it a random split?
- 4. **Normalization of your data.** Look at feature transformation and data normalization techniques, make your choices, and describe them in the paper.
- 5. My thoughts are to do batch normalization or normalize with sensible physical values (max value recordable via sensor).
  - 1. 1) Generally, data normalization is necessary for your model activation functions and to achieve good and stable learning.
  - 2. 2) You want to focus on the data's pattern, not the scale. I hypothesize that this will improve your model's ability to detect small avalanches or avalanches that are farther away.
  - 3. 3) You have different sensors at different geo-locations.
- 6. **Imbalanced data:** Explain how you address this (via sampling strategy?)
- 7. Data windowing: Are you adding additional features from larger window sizes? Such as mean, std, and frequency spectra to provide long-term context?
- 7. Model section: Please restructure the section. Make clear separations between the feature extractor and the evaluation task. Make it clear that you validate the autoencoders on a separate validation task. Separate our hyperparameter tuning. Etc. I recommend reading a couple of ML papers to get an intuition of how those sections are usually separated and written in a paper.
- 8. Model architecture: Please assign speaking names/identifiers to your three models that can be tracked throughout the complete paper (figures, text, abstract, appendix, tables) early on
  - 1. TAE: Why 1d convolutional layers?
  - 2. SAE: Could one use Fourier Neural Operators for this?
  - 3. Have you considered looking into methods to learn from imbalanced data? There are several models out there. Most commonly, people change their sampling strategy for their model to make sure the model sees positive and negative examples equally frequently.
- 9. Model learning: **Include learning curves** and show that the models are actually learning well. The reader wants to know if you are overfitting/underfitting. These figures can be included in the appendix, but they are essential to strengthening your paper. If your models are underfitting or not learning incredibly well, this would be a strong indicator that you can achieve even better results.
- 10. Model comparison: **Have more runs** (change random seed) to show whether differences are relevant/significant. Include error bars in your results. If you already run models across multiple seeds, report the number of runs and error bars.

**11. Figures**

- Each figure should bring one main point / finding across. The current figures show results, but they are not all structured or designed in a way that makes results easily accessible. A reader should be able to read a paper and access all main findings by reading the figures + captions.
- 2. Some figures need more descriptive captions.

**12. Discussion**

1. You need to back some of your claims (scalability, ability to generalize, etc.) with literature.

**13. Conclusion**

- 1. Adapt the sentence "We have shown that it bears strong potential..." (see comments). You have shown it can keep up with current state-of-the-art avalanche detection methods. Still, you have not demonstrated its potential (scalability, generalization ability, etc) in your experiments.
- 2. A more memorable last sentence talk about downstream applications (operational! Avalanche warning! You have such a strong and relevant use case, and your audience wants to hear about that.) would strengthen your conclusion.

**14. Code**

- 1. Looks good! Clean repo, nicely coded, well done.
- 2. You could add more documentation.
- 3. You have the code two times in the repo one seems to be for MacOS? Is this on purpose?
- 4. Mention the Python version in the Readme.
- 5. Add versions of your packages in requirements. Your code will not be reproducible later on if you do not report that (and it is not maintained).
- 6. Dir "models" and "lib" should be in "code" (semantically). I also get import errors if I do not move them over there.
- 7. Using your panda version broke the code for me (numpy pandas incompatability). I upgraded pandas to fix it.
- 8. Consider using Pathlib to handle path os-independently.
- 9. Add in your readme where people must change paths to get your code running on their machine.

**15. Data**

1. **Make sure to store the data on zenodo separately from the code.** You will want to update the data without updating the code in the future. If you want the data to be accessible (for other researchers and their projects), it needs to be maintained separately and with its own version control.

**Autoencoder-based feature extraction for the automatic detection of snow avalanches in seismic data**

Andri Simeon1, Cristina Pérez Guillén1, Michele Volpi2, Christine Seupel1, and Alec van Herwijnen1

**Correspondence:** Andri Simeon (andri.simeon@slf.ch)

[revised manuscript text omitted]
 lower-dimensional space, i.e. the latent space, which is designed and optimized to retrieve the relevant information of the given signal. For example, Mousavi et al. (2019) used an autoencoder to cluster seismic signals of an earthquake catalogue and showed comparable precision to supervised methods, while Kong et al. (2021) evaluated different

In this study, we explored at our study site above Davos (Sect. 2), Switzerland, throughout the winter seasons of 2020-2021 and 2021-2022. In Sect. 3, we determined the foundation of this dataset, which is one of the most critical parts of any machine learning model development. Similar to previous studies, we extracted features from 10 s seismic time windows and trained classifiers based on these features. In the feature extraction process (Sect. 4.1), we developed two new looks based on autoencoders, which learned to automatically extract 32 and 16 input features from the time and frequency domain respectively, and compared them against a set of 57 standard expert-based seismic attributes. The routines to optimize and train the autoencoder models are shown in Sect. 4.2. Using the different sets of input features, we trained three random forest classifiers to automatically distinguish the avalanche signals from other seismic events (Sect. 4.3). We analyzed and compared the performance of the models in Sect. 5. Finally, a discussion of the main results and conclusions are presented in Sect. 6 and 7.

autoencoder architectures for seismic event discrimination and phase picking.

**2 Study Site**

The study site is located at the end of the Dischma Valley, a tributary valley above Davos, Switzerland (Fig. 1). The seismic system was deployed on a flat meadow at about 2000 m a.s.l. (Eastern Swiss Alps; 46.72°N, 9.92°E). The surrounding mountains form a basin of steep slopes reaching up to 3000 m a.s.l. Since the winter season of 2020-2021, approximately from November to May, we installed a seismo-acoustic array of five co-located seismic and infrasound sensors arranged in a star-like pattern.

Figure 1. Left: Map and location of the study site. The instrumentation consisted of a seismo-acoustic array (blue dots), three cameras and a Doppler radar (red triangle). The approximate area where avalanches can be detected is shown for the seismo-acoustic array (blue ellipse) and the radar (red cone). Moreover, an avalanche path is highlighted with the red shaded area. Right: Photo taken by an automatic camera at the Dischma study site, showing the georeferenced path of a dry-snow avalanche released on 2 February 2022 at 02:31.

The seismic sensors were buried into the ground at a depth of approximately 50 cm and subsequently covered by snow during winter. A single measuring unit consists of a one-component seismometer Lennartz LE-1D/V (eigenfrequency of 1 Hz and sensitivity of 800 V m-1 s) and an infrasound sensor Item-prs (frequency response of 0.2-100 Hz and sensitivity of 400 mV Pa-1). The only exception is the central measuring unit applying a three-component seismometer LE-3Dlite (eigenfrequency of 1 Hz and sensitivity of 800 V m-1 s), of which we only used the vertical component in this study. The sensors were connected to the same digitizer (Centaur digitizer from Nanometrics), recording continuously with a sampling frequency of 200 Hz. The seismo-acoustic array monitors avalanches released from a per within a radius of approximately 3 km (blue ellipse in Fig. 1).

Additionally, the site is equipped with a Doppler radar and three automatic cameras to obtain independent validation data when weather conditions allow it, including accurate release times and information on the type and size of the avalanches. The radar emits electromagnetic waves that are reflected by the avalanche flow, providing the location and velocity of the moving avalanche (Meier et al., 2016). Figure 1 shows the location of the radar, which monitors several avalanche paths exposed to the west-southwest, covering an approximate area of 4 km² (red delineated area in Fig. 1). In the see, avalanches can be detected up to a maximum distance of approximately 2 km. The cameras automatically photograph every 30 minutes all the surrounding slopes (Fig. 1).

100

**3 Data**

115

135

We compiled a catalogue of seismic events from the continuous recordings of the winter season 2020-2021 and 2021-2022. Concretely, we manually picked events within periods of known avalanche activity and preprocessed the seismic signals. Then, three experts labelled the events, with which we finally compiled a two-class classification dataset.

[revised manuscript text omitted]

**3.3 Signal windowing and dataset splitting**

155

160

170

180

Before training the models, we further processed the event data in the catalogue. First, we treated the records of each seismic sensor independently yielding a five-fold enlement. Second, we applied a 10 s will window with 50% overlap to all signals. This windowing resulted in more data samples to train and ensured fixed-sized inputs for the models. Beyond, this strategy is also beneficial in a potential (near) real-time detection system, where 10 s windows are continuously parsed. With this, the labelled data set comprised 3'580 avalanche and 37'110 noise (non-avalanche) windows, which included 11'575 earthquake windows. This dataset is the foundation of this study and allows for systematic comparison of the methods in different settings.

Lastly, to develop the models and select the best architectures and hyper-parameters, we defined four independent data folds, i.e. three train folds for cross-validation and a test fold for assessing the performance on an independent inference set. We separated the folds by specific dates to prevent any correlation between the folds and reduce temporal data least. We chose the dates such that the class distributions across the folds are even (Fig. 3). The first train fold included dry avalanches exclusively, whereas the second contained a mixture of dry avalanches in the early part of the period, and wet avalanches in the latter. The third train fold and the fourth test fold spanned the winter season of 2021-2022. Again, the earlier counted towards dry conditions and the last both wet and dry.

**4 Model development**

**4.1 Feature extraction**

Feature extraction generally describes the compression of a signal to a lower dimensional embedding while retrievely preserving the signal's most distinctive information. The embedded information (the features) is usually into an upstream classification or regression task. Following this graph approach, we explore three methods to extract information from seismic signals either as learned feature vectors or domain-specific features, which are then classified as avalanche or noise.

In a first attempt, following a similar approach to Provost et al. (2017), which classified seismic events generated by landslides, we extracted a set of 57 predefined standard seismic attributes (Sect. 4.1.1). The feature engineering strategy is widely

Figure 3. Class distributions in the folds annotations on top of the bars depict the total number of 10 s seismic windows in each fold.

**Figure 4.** Overview of the three different approaches for avalanche classification. The blue elements depict the feature extraction, while the orange parts show the classification. Top (blue): The temporal autoencoder features; middle: The human-engineered seismic attributes; bottom: The spectral autoencoder features.

used in seismic detection of mass movements (Rubin et al., 2012; Provost et al., 2017; Lin et al., 2020; Wenner et al., 2021; Chmiel et al., 2021) and time series classification in general (Barandas et al., 2020). Additionally, it served as a benchmark for

comparing our second approach (Sect. 4.1.2), which is to learn the feature extraction completely unsupervised without making any preliminary assumptions about the signals. Using an unsupervised learning algorithm is beneficial when not provided with ground-truth labels, as in our case. Therefore, we used two autoencoder models to extract features from temporal and spectral input data, respectively (Sect. 4.1.2). The autoencoder concept was first introduced by Rumelhart et al. (1986) and has since been adapted for various applications (Lu et al., 2013; Mousavi et [1019; Gu et al., 2021). The architecture consists of an encoder and a decoder: The encoder compresses the input signal to a lower-dimensional embedding, i.e. the latent (feature) vectors. The decoder decompresses these feature vectors to the original input dimension. Overall, the autoencoder is trained by learning to reconstruct the input signals. Thus by design, the encoder feature vectors are optimized to preserve the most distinctive information characterising 
[revised manuscript text omitted]
  $\downarrow \equiv$  abilities, resulting in improved model  $\downarrow \equiv$  rmance (Fig. 9). The macro-average f1-scores increased by 2.6% (seismic attributes), 4.5% (TAE) and 5.4% (SAE). After ensembling, the seismic attribute and the SAE feature classification yielded similar performance in the classification metrics (see Table 2). Despite this improvement, the TAE feature classification still showed approximately double the liber of false alarms, i.e. 323 (14.7%), compared to the other models. The array-based aggregation further enabled us to investigate how predictions over an entire seismic signal evolve (Fig. 10). For the avalanche shown in Fig. 1 and 2, the models are comparably unsure in the starting

**Table 1.** Classification metrics on the (unseen) test fold data for the three feature sets. Due to the strong class imbalance, the weighted averages of the metrics are not shown.

| Model           | Class     | Precision | Recall | F1   | Support |   |
|-----------------|-----------|-----------|--------|------|---------|---|
|                 | Avalanche | 0.51      | 0.67   | 0.58 | 1335    | - |
| Seismic         | Noise     | 0.96      | 0.92   | 0.94 | 11135   |   |
| Attribute       | Macro Avg | 0.74      | 0.80   | 0.76 | 12470   | - |
|                 | Accuracy  |           |        | 0.90 |         |   |
|                 | Avalanche | 0.33      | 0.71   | 0.45 | 1335    | - |
| TAE             | Noise     | 0.96      | 0.83   | 0.89 | 11135   |   |
| Features        | Macro Avg | 0.64      | 0.77   | 0.67 | 12470   |   |
|                 | Accuracy  |           |        | 0.81 |         | _ |
|                 | Avalanche | 0.45      | 0.70   | 0.54 | 1335    | - |
| SAE
Features | Noise     | 0.96      | 0.90   | 0.93 | 11135   |   |
|                 | Macro Avg | 0.70      | 0.80   | 0.74 | 12470   | - |
|                 | Accuracy  |           |        | 0.87 |         |   |

Figure 8. La pace visualization of the most important features according to the impurity-based feature importance of random forest models for the seismic attributes (left), the temporal autoencoder features (middle) and the spectral autoencoder features (right). In parenthesis, the impurity-based importance of each feature is shown.

**Figure 9.** Results on the held-out test fold data after applying a probabilistic aggregation of the 10 s predictions over the 5 sensors of the array. The rows indicate the true (expert) labels, while the columns provide the predicted labels of the random forest classifiers. The colours code the percentage numbers.

phase, i.e. when it emerges from ckground noise. However, as the signal becomes more energetic, the avalanche probability increases for all models.

**Table 2.** Classification metrics on the (unseen) test fold data after probabilistic aggregation over the 5 sensors. Due to the strong class imbalance and bias towards the noise class, the weighted averages of the metrics are not shown.

| Model      | Class     | Precision | Recall | F1   | Support |
|------------|-----------|-----------|--------|------|---------|
|            | Avalanche | 0.56      | 0.68   | 0.61 | 267     |
| Seismic    | Noise     | 0.96      | 0.93   | 0.95 | 2202    |
| Attributes | Macro Avg | 0.76      | 0.81   | 0.78 | 2469    |
|            | Accuracy  |           |        | 0.91 |         |
|            | Avalanche | 0.37      | 0.71   | 0.49 | 267     |
| TAE        | Noise     | 0.96      | 0.85   | 0.90 | 2202    |
| Features   | Macro Avg | 0.67      | 0.78   | 0.70 | 2469    |
|            | Accuracy  |           |        | 0.84 |         |
|            | Avalanche | 0.53      | 0.71   | 0.60 | 267     |
| SAE        | Noise     | 0.96      | 0.92   | 0.94 | 2202    |
| Features   | Macro Avg | 0.75      | 0.82   | 0.77 | 2469    |
|            | Accuracy  |           |        | 0.90 |         |

**Figure 10.** Example of the seismic signal generated by an avalanche (up) and the mean output probabilities for each developed model over the entire avalanche signal (down). The probability is computed as the average of the individual probabilities predicted by each sensor every 5 seconds (10 s windows with 50% of overlap). The manual cuts are highlighted in dashed grey lines (upper plot), and the classification threshold 0.5 is in orange (lower plot).

**315 5.3 Event-based predictions**

Besides the single sensor and array-based predictions (Sect. 5.1 and 5.2), we investigated the predictions on the event level to close the gap to avalanche activity monitoring and provide a broader outlook. For this, we aggregated the array-level predictions in Fig. 9 over the entire duration of an event. We defined that a two consecutive windows (or 15 s of an event) had to be positively predicted for the entire event to be considered an avalanche. This threshold of two windows was not optimized. However, considering that the shortest avalanche in the dataset is 13 s, this boundary was feasible. This post-processing led to the results in Appen 3. Figure F1 shows a sill cant increase in avalanche recall with values of 0.82 (seismic attributes), 0.88 (TAE) and 0.91 (SAE). Nevertheless, the overall performance of the three models decreases by about 5% (see Table F4).

**6 Discussion**

320

325

So far, we compared the performance of a human-engineered seismic attribute classification and the autoencoder feature classification results based on a dataset containing 10 s seismic signals on a single sensor-level and multiple sensor-level (aggregation). With the latter aggregation, we observed a signi treduction in false alarms and a slight improvement in avalanche recall. Furthermore, we noticed that the automatically learned features, specifically the ones from the spectral autoencoder, the results showed that spectral input information seemed favourable. In the following, we contextualise the results by investigating the detection errors and their possible origins. Therefore, we summarize the model development (Sect. 6.1) and focus on the false predictions of the models to find potential limitations and

**Figure 11.** Array-based output probabilities of the random forest models for their respective input features with expert avalanche scores. The blue dashed line indicates the threshold applied to the expert scores to assign avalanche class labels.

reasons (Sect. 6.2 and 6.4). Finally, we argue about the applicability of these models (Sect. 6.3) and compare the results to previous work (Sect. 6.5).

**6.1 Model performance and limitations**

335

340

345

350

Machine-learning models are strongly influenced by the quality and size of the dataset. The relative mall size constrained us to design autoencoder architectures with rather few trainable weights. In addition, we used each sensor independently to compensate for dataset size, as each sensor can be considered a different view of the same event. However, this came at the cost of introducing correlation among dataset samples as the sensors were installed nearby (Fig. 1) and thus recorded very similar signals, yet not necessarily adding much new and enriching information to the dataset. Given that the dataset will increase in the next years, we will consider incorporating the 5 sensors as distinct channels in a convolute of model in future studies. With this, the sensor aggregation and fusion would be implicitly implemented into the model. Another aspect to bear in mind was the signal normalization. Normalizing input data has proven crucial when training neural networks (Sola and Sevilla, 1997). The temporal autoencoder, in particular, therefore loses information on absolute and relative amplitudes. Yet, both autoencoders ould still capture signal characteristics and remarkably show similar patterns when looking at continuous predictions (see Fig. 10). Alternatively, a normalization over the entire signal before windowing could be envisioned to preserve information on relative amplitudes. However, this is not practical for (near) real-time signal detection.

Further, the dataset drove the decision to separate the feature extraction and classification. The unsupervised feature extraction is not constrained to a labelled dataset (only the model selection and hyper-parameter tuning of the classifiers are), an advantage when dealing with non-ground-truth labels (two-thirds of the avalanches were not verified). This allowed us to analyze a lower-dimensional embedding of the dataset by inspecting the feature space distributions (Fig. 8). As labels for earthquakes were available, we visualised the earthquake class separately. Moreover, earthquake and avalanche signals can be

similar in the time domain (Heck et al., 2019a), thus we wanted to investigate them in the feature domain. In an early stage, we trained models with three classes (earthquake separately), without seeing an increase in overall model performance. In addition, note that training a model to also classify earthquakes was out of scope as these can be detected with other methods. Overall, the three event types, i.e. avalanches, earthquakes and rest, varied in the encoding locations, yet also showed considerable overlap. Interestingly though, the avalanche and earthquake signals were well separated (blue and orange in Fig. 8). The rest (grey) resembled a connecting cloud between avalanche and earthquake signals. The reason for this might be two-fold; first, the heterogeneity of these noise events by potentially comprising minor avalanches and low magnitude earthquakes (false negatives), and second, the strong attenuation in some sections of avalanche signals resulting in low amplitude avalanche windows. The heterogeneity within the noise class originated from including different sources in comparable amplitude ranges, e.g. earthquakes, aeroplanes or strong wind. However, these various sources are definitive to be expected and need to be considered in a real-time detection system.

In future implementations, further investigations could also be conducted considering the avalanche class by differentiating between type and sizes. Since the primary goal of this study was to develop and compare models to detect avalanches regardless of their type or size, we trained the models considering all the recorded avalanches. Therefore, we ensured that various avalanche types were included in both the train and test set by separating them based on appropriate dates (Sect. 3.3). According to radar and image data, most avalanches detected by our seismic array ranged between size classes 2 and 3, based on the European avalanche size classification (EAWS, 2021). Future models could be expanded to also classify avalanches by size and type. Given that seismic patterns of avalanches are influenced by the avalanche type (Pérez-Guillén et al., 2016), an alternative approach could be to develop two independent models to detect dry-snow and wet-snow avalanches separately. However, the current dataset was too small to further categorize the avalanche events by size and type, and accurate ground-truth data was often also missing.

Finally, the applied expert labelling was subject to an unknown degree of subjectivity and belief for the non-verified events. In addition, having decided upon a hard threshold to convert expert scores to class labels further blurred the boundaries between the avalanche and noise class, i.e. the noise class might include minor avalanches (false negatives). We, therefore, investigated the relationship between the random forest's output probabilities and the expert scores of potential avalanche signals (Fig. 11). Also, we found the average expert agreement rate on the avalanche samples to be 58%, i.e. on average, two experts agree on 58% of the avalanche signals. Overall, the output probabilities of the random forest models positively increased with the expert scores. As expected, we also noted the highest uncertainty at the selected threshold (dotted blue line the comparing the feature sets, the classification with the seismic attributes yielded clearer steps over expert scores and more distinctive probabilities for the highest and lowest expert scores. A measure to mitigate having to deal with noisy labels in future works might be to solely include verified avalanches and discard the non-verified ones for training the models. However, the unsupervised autoencoders are entirely independent of any class labels or information. Thus, by considering only verified avalanches, we would not reduce class ambiguity from the autoencoder's perspective but the dataset size and with it, valuable information might be lost.

[revised manuscript text omitted]

Overall, our results thus showed that using an array of sensors helped to reduce the number of false avalanche detections by averaging the predictions of the sensors. This can be viewed as model ensembling and is generally known to improve results (Mohammed and Kora, 2023). Secon cluding features from the frequency domain tended to show fewer FPs. Third, an interesting and positive finding was that the models rarely confused earthquakes for avalanches (on average 4.3% of all earthquake windows). Moreover, the models generate false alerts to a similar extent to previous studies in avalanche detection (e.g. Bessason et al., 2007; Rubin et al., 2012; Hammer et al., 2017; Heck et al., 2018). Thus, they might not yet be suited for an early-warning application. However, the models could be implemented in an avalanche activity assessment process or to label unverified events in the future by being aware of the limitations and that they tend to produce too many avalanche detections. In pursuit of reducing the number of false alerts, one might consider including other types of recordings, e.g. infrasound data (Mayer et al., 2020). Also, implementing specialized data augmentation techniques to increase the variety and number of the avalanche recordings, e.g. seismic data augmentation techniques (Zhu et al., 2020) or generative models (Wang et al., 2021) might help to make the classifiers more robust to changing environments and setups. Classifier robustness is another compelling prerequisite when considering the transferability of such models to other test sites and shoul considered in future studies. We would expect variations in the detection performance to arise from different configurations in the study site setup, sensor location and configuration as well as in the characteristics of the terrain and the avalanches.

**6.5 Comparison to previous studies**

460

465

470

475

480

485

To conclude, we put our results in a broader context by comparing them with previous studies. Provost et al. (2017) used a random forest model based on the 71 engineered seismic attributes. They reported stunning true positive rates of 94%, 93% and 94% for the rockfall, quake and earthquake class and a true negative rate of 92% for the noise class. The setting, however, is difficult to compare, as they used non-windowed signals from an evenly distributed dataset comprising 418 rockfalls, 239 quakes, 407 earthquakes, and 395 noise events.  $\blacksquare$ , these event types typically generate signals with a higher signal-to-noise ratio than avalanches. Moreover, they included polarity and network attributes in the features, which for the classification turned out to be most important. Nevertheless, with 92% true negative ir model is comparably prone to producing false alerts as the models in this study are. Also, for avalanche detection, several studies presented the approach of feature engineering and subsequent classification (e.g. Bessason et al., 2007; Rubin et al., 2012; Hammer et al., 2017; Heck et al., 2018). Rubin et al. (2012) used 10 engineered features in the frequency domain and tested 12 classification models, of which the decision stump classifier showed the highest overall accuracy of 0.93. However, the model showed a poor precision of 0.13, hence, producing many more false alerts. In contrast to our approach, they only considered avalanches verified on camera images or manually picked events. Heck et al. (2018) used the same avalanche catalogue of 283 avalanches, of which 25 were confirmed and the rest were labelled by three experts. They implemented engineered temporal and spectral features and used an HMM as a classifier. Similar to most previous studies, they also noted high values of FPs. Moreover, they observed improvements when aggregating single sensor to array-based predictions as we did in this study. In conclusion, based on the results of this and previous studies, we expect that an avalanche predictor based on solely seismic data will always produce false alarms, as it remains a difficult task to identify low-energy avalanche signals. Therefore, installing a secondary seismic detection system in the proximity of the avalanche path would be advantageous in mitigating false alarms. Alternatively, integrating a complementary detection system, such as an infrasound system, could also be beneficial but less cost-effective.

In summary, the classification results met the performance of previous studies on avalable he detection. However, the core contribution of this study is two alternatives to extract features from seismic signals. We showed that the proposed encoder features are applicable for avalanche detection and compare well to engineered features. In particular, the learned feature extraction does not depend on prior expertise or knowledge and thus can be adapted easily to new settings, e.g. changing environments, without having to set some parametrisations of expert features. Moreover, with growing dataset size or larger datasets, it can improve over time. Finally, a future interesting comparison would be to evaluate the models on how they generalize to other test sites and settings.

**7 Conclusions**

We proposed two unsupervised seismic feature extraction methods based on deep learning algorithms and a set of seismic attributes to train three random forest classifiers for avalanche detection. The dataset was compiled from seismic avalanche data recorded during two winter seasons in Davos, Switzerland. While in earlier studies, seismic data classification

mostly followed the approach of extracting well-defined signal attributes to train classifiers, the proposed deep learning models

bridge the gap to a purely learned (automatic) pipeline.

Overall, the classifiers achieved macro-average f1-solved ranging from 0.70 to 0.78 with avalanche recall values ranging from 0.68 to 0.71. Moreover, the results clearly show that including features from the frequency domain improves model performance. Further, as we observed that the onset and end of avalanche signals were often misclassified as noise but the most energetic signal parts were not, we proposed a simple post-processing step. By imposing that at least two consecutive prediction windows, i.e. 15 s, must be positive for an entire event to be positive, we drastically reduced the missed avalanches (false negatives). This criterion significantly improves the avalanche recall, ranging from 0.82 to 0.91. Lastly, contrary to previous expectations, earthquakes are rarely mistaken for avalanches at our study site.

495

500

Revisiting our primary goal of comparing human-engineered with automatic feature extraction, there is enying that the standard expert-based seismic attributes classification is a robust approach. These predefined attributes have been studied and applied for a decade and optimized and tuned throughout various studies. The unsupervised representation learning, in contrast, is a completely new approach to seismic avalanche data analysis. We have shown that it bears strong potential for future implementations and applications. Compared to the engineer features, the learned features require no prior expertise and, therefore, can easily be adapted to changing environments without having to set some parametrisations of expert features. Also, they can improve with growing dataset size and quality in future.

Code and data availability. The code and data are available on Zenodo (doi: 10.5281/zenodo.12162570). It is predominately written in Python using the PyTorch library (Paszke et al., 2019) for the autoencoder design, the random forest implementation of the Scikit-learn library (Pedregosa et al., 2011), the Pandas library (Wes McKinney, 2010) for handling the data and more standard Python libraries such as NumPy (Harris et al., 2020) and SciPy (Virtanen et al., 2020).

I would consider moving such descriptions to a section called "Experimental Setup" or similar. If GMD encourages to mention this in that section, you should keep it of course.

[revised manuscript text omitted]

  - EAWS: European Avalanche Size Scale, https://www.avalanches.org/standards/avalanche-size/, [Online; last access 3-September-2024], 2021.
- Gu, S., Kelly, B., and Xiu, D.: Autoencoder asset pricing models, Journal of Econometrics, 222, 429–450, https://doi.org/10.1016/j.jeconom.2020.07.009, 2021.
  - Hammer, C., Ohrnberger, M., and Fäh, D.: Classifying seismic waveforms from scratch: a case study in the alpine environment, Geophysical Journal International, 192, 425–439, https://doi.org/10.1093/gji/ggs036, 2013.
  - Hammer, C., Fäh, D., and Ohrnberger, M.: Automatic detection of wet-snow avalanche seismic signals, Natural Hazards, 86, 601–618, https://doi.org/10.1007/s11069-016-2707-0, 2017.

- Harris, C. R., Millman, K. J., van der Walt, S. J., Gommers, R., Virtanen, P., Cournapeau, D., Wieser, E., Taylor, J., Berg, S., Smith, N. J., Kern, R., Picus, M., Hoyer, S., van Kerkwijk, M. H., Brett, M., Haldane, A., del Río, J. F., Wiebe, M., Peterson, P., Gérard-Marchant, P., Sheppard, K., Reddy, T., Weckesser, W., Abbasi, H., Gohlke, C., and Oliphant, T. E.: Array programming with NumPy, Nature, 585, 357–362, https://doi.org/10.1038/s41586-020-2649-2, 2020.

[revised manuscript text omitted]

  - Virtanen, P., Gommers, R., Oliphant, T. E., Haberland, M., Reddy, T., Cournapeau, D., Burovski, E., Peterson, P., Weckesser, W., Bright, J., van der Walt, S. J., Brett, M., Wilson, J., Millman, K. J., Mayorov, N., Nelson, A. R. J., Jones, E., Kern, R., Larson, E., Carey, C. J., Polat, İ., Feng, Y., Moore, E. W., VanderPlas, J., Laxalde, D., Perktold, J., Cimrman, R., Henriksen, I., Quintero, E. A., Harris, C. R., Archibald, A. M., Ribeiro, A. H., Pedregosa, F., van Mulbregt, P., and SciPy 1.0 Contributors: SciPy 1.0: Fundamental Algorithms for Scientific Computing in Python, Nature Methods, 17, 261–272, https://doi.org/10.1038/s41592-019-0686-2, 2020.
  - Wang, T., Trugman, D., and Lin, Y.: SeismoGen: Seismic Waveform Synthesis Using GAN With Application to Seismic Data Augmentation, Journal of Geophysical Research: Solid Earth, 126, e2020JB020077, https://doi.org/https://doi.org/10.1029/2020JB020077, e2020JB020077 2020JB020077, 2021.
- Wenner, M., Hibert, C., Herwijnen, A. V., Meier, L., and Walter, F.: Near-real-time automated classification of seismic signals of slope failures with continuous random forests, Natural Hazards and Earth System Sciences, 21, 339–361, https://doi.org/10.5194/nhess-21-339-2021, 2021.

[revised manuscript text omitted]

**Table F2.** Summary of the TAE hyper-parameter optimization. Shown are only the models for which all three metrics are ranked in the top 20. The best metrics and the selected model are highlighted in bold.

| Weights | Filters in first Layer | Feature
Dimension | Kernel
Size | Stride | Expected Avalanche Portion | Augmentation | Silhouette
Score | Calinski–Harabasz
Index | MSE   |
|---------|------------------------|----------------------|----------------|--------|----------------------------|--------------|---------------------|----------------------------|-------|
| 109865  | 8                      | 64                   | 8              | 4      | default                    | False        | 0.191               | 849.959                    | 0.078 |
| 109865  | 8                      | 64                   | 8              | 4      | 0.5                        | False        | 0.024               | 357.494                    | 0.073 |
| 109865  | 8                      | 64                   | 8              | 4      | 0.5                        | True         | 0.018               | 345.684                    | 0.076 |
| 156945  | 16                     | 32                   | 20             | 10     | 0.5                        | False        | 0.033               | 374.174                    | 0.06  |
| 156945  | 16                     | 32                   | 20             | 10     | 0.5                        | True         | 0.011               | 567.276                    | 0.055 |
| 514337  | 32                     | 32                   | 20             | 10     | default                    | True         | -0.072              | 368.876                    | 0.054 |
| 514337  | 32                     | 32                   | 20             | 10     | 0.5                        | False        | 0.061               | 333.174                    | 0.061 |
| 514337  | 32                     | 32                   | 20             | 10     | 0.5                        | True         | 0.041               | 613.917                    | 0.054 |
| 625185  | 32                     | 64                   | 20             | 10     | 0.5                        | False        | -0.095              | 292.78                     | 0.063 |
| 625185  | 32                     | 64                   | 20             | 10     | 0.5                        | True         | -0.105              | 307.477                    | 0.064 |

**Table F3.** Summary of the SAE hyper-parameter optimization. Shown are only the models for which all three metrics are ranked in the top 10. A "default" hidden dimension indicates that the dimensions in the layers of the encoder linearly decrease from the input dimension (200) to the feature dimension. The best clustering metrics and the selected model are highlighted in bold.

| Weights | Layers | Feature
Dimension | Activation
Function | Hidden
Dimensions | Silhouette
Score | Calinski–Harabasz
Index | MSE   |
|---------|--------|----------------------|------------------------|----------------------|---------------------|----------------------------|-------|
| 47552   | 2      | 16                   | Tanh                   | default              | 0.227               | 1205.952                   | 0.014 |
| 47552   | 2      | 16                   | leaky ReLU             | default              | 0.218               | 1088.234                   | 0.012 |
| 70880   | 2      | 64                   | Tanh                   | default              | 0.198               | 999.475                    | 0.014 |
| 81330   | 3      | 16                   | Tanh                   | default              | 0.224               | 1237.579                   | 0.013 |
| 81330   | 3      | 16                   | leaky ReLU             | default              | 0.217               | 1015.357                   | 0.012 |
| 112432  | 4      | 16                   | Tanh                   | default              | 0.238               | 1111.027                   | 0.013 |
| 112432  | 4      | 16                   | leaky ReLU             | default              | 0.223               | 1013.013                   | 0.012 |
| 146120  | 5      | 16                   | leaky ReLU             | default              | 0.223               | 968.953                    | 0.012 |

**Table F4.** Classification metrics on the (unseen) test fold data after the aggregation over entire events of the array-based predictions. Due to the strong class imbalance and bias towards the noise class, the weighted averages of the metrics are not shown.

0.7

0.87

0.73

0.85

308

Macro Avg

Accuracy

---

## Referee Report (RR4)

Autoencoder-based feature extraction for the automatic detection of snow avalanches in seismic data

Andri Simeon et al.

**Referee Comment**

This manuscript develops autoencoder derived seismic attributes and engineered seismic attributes as features in a random forest classification detection of snow avalanches. The results suggest that the autoencoder derived attributes perform as well as the engineered seismic attributes for event detection. The avalanche detection method is reported to be potentially used as an operational, near real-time avalanche detection system, though the relatively high number of false alarms requires further improvement. The work is presented with respect to the previous studies employing machine learning for seismic event detection while highlighting their significant and novel contribution of unsupervised feature extraction. I found the paper to be informative and complete in analysis and have been satisfied by the authors' responses to the initial review which improved the methodological development and comprehensibility of the work.

**Comments:**

Line 6: "Therefore, we compiled a dataset of seismograms recorded with an array of five seismometers..." This sentence seems a bit disconnected from the main ideas presented. Is this statement intended to link back to "Monitoring snow avalanche activity is essential for operational avalanche forecasting..." or "Still, automatically distinguishing avalanche signals from other sources in seismic data remains challenging." I think the intent could be clarified by replacing "Therefore" with a descriptive intro to sentence like, "Because of the inherent complexity of interpreting signals traveling within the subsurface, we utilized an array of five seismometers..." This example expresses the importance of having an array of seismometers.

Line 104 - 106: Feels like a run-on sentence. Consider the revision, "Additionally, the site was equipped with a Doppler radar and three automatic cameras to obtain independent validation data, including accurate release times and information on the type and size of avalanches, provided favorable weather conditions."

Line 109- 110: "The cameras automatically photographed all surrounding slopes every 30 minutes (Fig. 1)." Consider including one sentence detailing how the photographs were utilized. Manually inspected as a corroboratory inspection of radar or other data source detections, or automatically reviewed as an independent method? This is mentioned on lines 113-114, but not explicitly.

Line 132: What is exactly meant by ground velocity? The derivative of the seismic displacement? Just a bit more detail on this method would be helpful.

Line 232: "As activation function" → "As an activation function"

Line 282: "Therefore, we used the three train folds"  $\rightarrow$  "Therefore, we used the three training folds"

Line 306: Similarly "train" → "training"

Line 307: "weigh"  $\rightarrow$  "weight"

Line 465 "Tough" → "Though"

One thing to note: Between the Engineered Feature and AE feature avalanche detection, which had comparable recall values, were the same avalanches detected? Could the implementation of both SAE and engineered feature detection further increase the detection capability. For if they detected the same amount of avalanches, but different ones, perhaps this increases the overall detection. This analysis is explicitly missing, but could provide additional insights to the differences of the detection methods. Perhaps this is something you have already investigated, but did not note in the manuscript. Figure 13 touches on this conceptually, but it is still hard to discern if the false positive detections stem from different events or not.

With Regards,

Tate Meehan

---

## Author Response (AR2)

Dear editors, Dear reviewers,

We would like to express our deep gratitude for the outstanding effort made by the reviewers in providing feedback, which has greatly improved our manuscript. With the submission of the revised manuscript, we hope to meet the high standards requested.

Kind regards, Andri Simeon

**Table of Contents**

| REFEREE 3#:       |    |
|-------------------|----|
| GENERAL COMMENTS  | 2  |
| SPECIFIC COMMENTS |    |
| REFEREE 4#:       | 6  |
| GENERAL COMMENTS  | 6  |
| SPECIFIC COMMENTS | 6  |
| REFEREE 5#:       | 8  |
| GENERAL COMMENTS  | 8  |
| SPECIFIC COMMENTS | 8  |
| REFEREE 6#:       | 13 |
| GENERAL COMMENTS  | 13 |
| SPECIFIC COMMENTS | 15 |

**Referee 3#:**

**General Comments**

This study develops deep-learning models to identify seismic signals of avalanches. I am a seismologist working on non-earthquake signals (avalanches, landslides, glaciers...).

But I have no expertise on deep learning methods.

The introduction and the seismological methods are well described. But I had trouble to understand the deep-learning methods.

I think the authors did not make much effort to make their work understandable and interesting to readers that are not familiar with deep learning methods.

I did not know many terms (f1 score, recall ...), but I learned from other sources. But readers of GMD are probably more familiar with deep learning methods than me. I am thus not able to criticize the part on deep leaning methods.

The results of the model seem correct, but not as good as previous similar studies (eg, Provost et al), and maybe not good enough for real time warning.

We thank the reviewer for providing valuable feedback on our work. Introducing and explaining deep learning building blocks in detail is beyond the scope of this work. The neural network layers and metrics are state-of-the-art and well-documented. Excellent online resources exist for readers who wish to delve deeper into the underlying mechanisms. Nevertheless, we have revised the manuscript in accordance with all the reviewers' comments and enhanced the description of the machine learning models for improved clarity.

Considering the previous work of Provost et al., 2017, we would like to emphasise that a direct comparison is challenging since their objective is entirely different. They developed a model to detect entire events rather than subsequences. In this study, we adopted their feature extraction method in our baseline to make a comparison possible. However, the signals generated by landslides and avalanches are different. Landslides usually generate higher amplitudes and frequency content, which depend on the sensor-source distance. In addition, the signals generated by landslides are usually longer than avalanche signals. For instance, Provost et al. (2017) used different frequency bands to compute the seismic features and used a cross-validation strategy to evaluate the model's performance instead of using an independent test set.

**Reference:**

Provost, F., Hibert, C., and Malet, J. P.: Automatic classification of endogenous landslide seismicity using the Random Forest supervised classifier, Geophysical Research Letters, 44, 113–120, https://doi.org/10.1002/2016GL070709, 2017.

**Specific Comments**

**1) Objectives**

It is not clear to me what are the long-term objective of this work, beyond showing that machine learning can detect seismic signals of avalanches, more or less as well as experts? If the final goal is to monitor avalanches and investigate their flow properties, experts could do that better and faster than machines, given the relatively small rate of avalanches. If the goal is real-time warning, do you think that there would be enough warning time (from detection until propagation to roads, villages ...) to allow mitigation actions? We have tried to make our objectives more transparent in the introduction (Lines 72 -76). Monitoring avalanches on a large scale in (near) real-time is a long-term objective of this study, which is not feasible to carry out manually. Avalanches typically occur during winter storms, making it impossible for humans to monitor releases, especially in remote areas. Also, automatic cameras cannot help in these cases. Therefore, expensive Doppler radars are the most reliable solution to monitor a single avalanche slope. In contrast, seismic detection systems provide a cost-effective solution that is scalable to cover wider areas and, therefore, offers a higher spatial resolution of avalanche activity. But then again, having experts interpret all this data in real-time is not practical. Thus, developing reliable algorithms to automatically classify signals in seismic data is a necessary first step before setting up an early warning system. Hence, we studied and developed models that could provide a solution in the future. A question that naturally appears when trying to monitor avalanche activity in (near) real-time is that of real-time warning. However, since developing a real-time warning is a desirable future application of this study, we presented and discussed this as a broader outlook in the Section

**2) Time window**

Seismic signal is divided in windows of 10 s, much smaller than the typical duration of avalanche seismic signals.

«Applicability to early warning and monitoring systems» (Section 6.4).

I understand that a short window is preferable for early waring (but is this your goal and is it feasible?).

But for classification, I think a longer time window (about 60 s) would improve the results by allowing to better detect the characteristic spindle space of avalanche signals and the consecutive P, S and coda waves of earthquakes

A shorter time window is preferable for early warning and real-time monitoring of avalanche releases. Most importantly, using small window sizes enlarged the avalanche data used to train the models. Additionally, working with smaller time windows would, in principle, allow for the segmentation of different events, hence capturing the duration, release and setting of avalanches and offering additional data to study events. Developing a neural network from scratch using only 84 avalanche signals, i.e., the number of recorded avalanches, is unrealistic. However, we agree that longer windows would better capture the characteristic shapes of different signals, and we will consider this for future work.

**3) Network based attribute**

Why did you choose not to use network attributes, although you have a network of 5 sensors and network attributes were found to improve the results in the study of Provost et al. (2017)? For instance, computing cross-correlation between sensors provides the peak correlation and time delays, which could be useful to distinguish avalanches from earthquakes or noise sources.

The correlation between sensors is much larger for distant sources (earthquakes, quarry blasts) than for nearby signals (avalanches and noise).

The time delay depends on apparent velocity, which is larger for deep and distant sources (earthquakes) than shallow and nearby source (avalanches, acoustic waves).

We avoided using network attributes since we wanted to explore whether a single sensor suffices to detect avalanches. Our reasoning was based on anticipating future scenarios where having multiple sensors cannot be taken for granted when moving to large-scale avalanche monitoring. Again, like the windowing, using the five sensors separately provided us with more data to develop the models. In practice, when multiple sensors are available, their integration, as well as derived data, can be seamlessly incorporated into the system.

**4) Infrasound**

Each seismic sensors was colocated with an infrasound sensor. Why didn't you use infrasound data?

I guess that the amplitude of the infrasound signal (relative to the seismic signal) should be much higher for avalanches than for earthquakes?

Incorporating infrasound data was not within the scope of this study. Furthermore, the acoustic system did not function correctly during the first winter season, that is, 2020-2021.

Consequently, the database available for developing machine learning models was too small. Nevertheless, we agree that this system could provide valuable additional insights and will be considered for future work.

**Minor comments:**

Figure 8: what are the main features shown in x axes? ("ES[3]" "DISTQ3Q1" "F21"...?).

We have added a brief description of the three engineered features in the caption. ES[3] is the energy in the frequency band [6, 9] Hz, DISTQ3Q1 is the mean distance between the 3rd and the 1st quartile of all FFTs as a function of time and DISTQ3Q2 is the mean distance between the 3rd and the 2nd quartile (Feature 36, 57 and 56 in Table B2 and B3). The features starting with 'F' are the features from the autoencoders. They do not have any direct physical meaning.

Figures 2 and 12: Could you also show the spectrogram of the signal? Avalanches are often easier to identify by looking at the spectrograms than seismograms.

We have added all spectrograms alongside the waveforms in the revised version.

l349 "two-thirds of the avalanches were not verified". You mean, by cameras or radar? But they were verified by experts?

One-third has been verified by radar and/or cameras, so it can be considered ground truth. The remainder was labelled by experts and, thus, are not verified ground truth. We clarified this by adapting the text to (Line 404):

... (two-thirds of the avalanches were neither verified by the radar nor the cameras).

l424-425 What do you mean by "Thus, the models could, in turn, be considered to annotate large datasets, which in turn can be used to detect fine precursor signals."?

We realised this sentence was not clear and informative. Therefore, we have removed it from the manuscript. Initially, we wanted to say that the models could label new or unlabelled datasets. These datasets could then be used to develop new models to detect the initial signals of an avalanche for early warning.

l462 "Also, these event types typically generate signals with a higher signal-to-noise ratio than avalanches".

I don't agree. All gravitational and tectonic processes generate a majority of small events (eg, Gutenberg Richter law for earthquakes). The minimum size is always the detection threshold. We have removed it from the manuscript.

l469 "In contrast to our approach, they only considered avalanches verified on camera images or manually 470 picked events."

I don't understand. I thought you also manually picked the avalanche seismic signals and verified part of them with radar and cameras?

We have removed it from the manuscript.

**Referee 4#:**

**General Comments**

This manuscript develops unsupervised machine learning techniques of seismic data recordings to detect snow avalanches, showing a success level greater than expert interpretations and providing insights and future improvements in the study. The work is presented with respect to the previous studies employing machine learning for seismic event detection while highlighting their significant and novel contribution of unsupervised feature extraction. I found the paper to be informative and complete in analysis yet lacking full clarification and description of key methodological steps. I recommend the paper for publication after minor, clarifying revisions, which will improve the readability and methodological completeness of the manuscript. We appreciate the time and effort that the reviewer has invested in providing valuable feedback on our work. We have thoughtfully considered the points raised and have endeavoured to improve accordingly. Our responses to the reviewer's suggestions are detailed below.

**Specific Comments**

Section 4 is difficult to wade through with lengthy explanations which leave out important methodological developments.

Reviewer #6 had similar concerns. Therefore, we put a lot of effort into rewriting and restructuring the method section. For details on this section, we refer to the marked-up manuscript, which contains all the changes.

Section 4 is now structured as follows:

```
(4) Model development
```

(4.1) Baseline features

(4.2) Autoencoder features

(4.2.1) Architecture

(4.2.2) Training Regime

(4.2.3) Validation

(4.2.4) Model selection

(4.3) Feature classification

(4.3.1) Random forest model

(4.3.2) Cross-validation

(4.3.3) Inference and post-processing

Section 4.3 contains a significant amount of information which reads as discussion material and should be placed into Section 6 to improve the readability of the manuscript. Paragraphs have been highlighted in the marked-up document to convey this.

We have combined and relocated both of these paragraphs to the discussion section. This should enhance the reading flow and minimise duplicated text.

Line 281 "During inference, each tree prediction is aggregated to form a final majority vote, from which it is possible to retrieve class proportions, often interpreted as probabilities."

The statement on Line 281 contains the entirety of the description of the model output. More care and consideration should be taken to clarify exactly how random forest classification is interpretable as a probability. It should also be stated why 0.5 probability was chosen as the threshold for event detection. We acknowledge that the actual model output was not described in full detail. Consequently, we have taken this comment into account and added a subsection titled «Inference and post-processing» to the section «Feature classification». This subsection explains how the random forest model arrives at its predictions and what steps we have taken to post-process them.

The authors should refine the work to place methodologies in the appropriate sections.

The process of multi-sensor data aggregation is in introduced within the results on line 306, and vaguely at that. This idea can be introduced in Section 4. Additional examples of this have been identified in the marked-up document. To clarify which tasks we evaluated our models on, we have added the aforementioned «Inference and post-processing» section, which introduces all the aggregations we utilised.

Figures in Section 4 are lacking the necessary detail to be supportive/educational information. The authors should at a minimum improve the captions to better convey the depictions.

Thank you for this suggestion, the captions were indeed relatively sparse. Therefore, we have added more descriptive captions to Figure 5 (temporal autoencoder) and Figure 6 (spectral autoencoder). Additionally, taking into account another reviewer's feedback, we redesigned the entire Figure 4.

Additional Comments:
See the marked-up document included.

With Regards, Tate Meehan

**Referee 5#:**

**General Comments**

The manuscript by Simeon et al. presents a novel method to automatically detect snow avalanches in seismic data collected at a test site above Davos, Switzerland. Specifically, the performance of three different algorithms is assessed: seismic attributes, temporal autoencoder, and spectral autoencoder. Based on this, they find that the inclusion of features from the frequency domain improves model performance and unsupervised autoencoders show potential as an alternative to the standard expert-based seismic attributes classification.

I believe the science behind this study is sound and aligns with the focus of Geoscientific Model Development. However, I suggest some restructuring and clarification prior to publication. In particular, the Discussion section is quite long and, as pointed out by another referee, the manuscript still contains some repetitive information. Considering Provost et al. (2017) found higher true positive rates for non-windowed signals, I believe determining the effect of different window lengths would add value to the manuscript. At least the potential effects of choosing a different window length should be discussed. I recommend the authors also take my specific comments listed below into account.

Thank you very much for reviewing our work and providing insightful feedback. Since other reviewers also suggested some restructuring, we have improved the readability of the paper by reorganising the methods and discussion sections. In parallel, we have removed duplicative information where we thought it was not needed or explicitly highlighted by a reviewer. We believe that some level of repetition is essential to provide context for related information. For the responses to the specific comments, we refer to the list below.

**Specific Comments**

L41 – radius of several kilometres: This becomes clearer later on in the manuscript, but I would suggest being precise from the very beginning.

This definition should provide the reader with a vague idea of the detection radius of such systems. Seismic waves are rapidly attenuated with distance, giving rise to a natural limit of detection of signals produced by avalanches. The detection range of a seismic network varies depending on the type and size of the flow, the characteristics of the terrain, and the background noise level. Hammer et al. (2017) used a seismic station of the Swiss seismological network to detect large wet-snow avalanches up to 30 km. Pérez-Guillén et al. (2019) detected large avalanches released on Mt. Fuji (Japan) at a maximum distance of 15 km. Therefore, making a quantitative statement on the detection radius that holds in general is difficult.

**References:**

Hammer, C., Fäh, D., and Ohrnberger, M.: Automatic detection of wet-snow avalanche seismic signals, Natural Hazards, 86, 601–618, https://doi.org/10.1007/s11069-016-2707-0, 2017

Pérez-Guillén, C., Tsunematsu, K., Nishimura, K., and Issler, D.: Seismic location and tracking of snow avalanches and slush flows on Mt. Fuji, Japan, Earth Surface Dynamics, 7, 989–1007, https://doi.org/10.5194/esurf-7-989-2019, 2019.

L46 – other types of mass movements: Consider adding a few examples.

We have added:

... such as landslides, debris flows, and lahars.

**L47 – other seismic sources "such as earthquakes"?**

We have changed this sentence to:

These patterns have frequently been used to detect and identify avalanche signals.

**L71: You describe what an encoder is, but not a decoder.**

Following another reviewer's suggestion and in order to avoid duplication, we have removed the detailed description of the autoencoder model from the introduction. Nevertheless, thank you for noting. A thorough introduction to the autoencoder architecture can be found in the methods section.

**L80: Why did you use 10 s windows and not, e.g., 5 s, 20 s, or maximum length of picked avalanche events?**

Referee #3 expressed a similar concern. In summary, by using smaller windows, we could generate more avalanche samples, which were needed to develop the autoencoders. Moreover, a short time window is preferable for potential real-time applications. Thus, 10 seconds is a compromise between compiling a larger number of avalanche samples for training and avoiding a reduction in length to a point where the avalanche is not detectable anymore. Given that the shortest avalanche recorded at our site was 13 seconds, we thought empirically that this was a good choice. Nonetheless, we also intend to further analyse this in future work, given the number of avalanche samples increases.

**L91: What are the advantages of a star-like pattern compared to others?**

This is used to localise avalanches through array processing methods. To clarify this we have added the following sentence:

This spatial configuration allows for the localization of avalanches (Heck et al., 2018a).

**Reference:**

Heck, M., Hobiger, M., van Herwijnen, A., Schweizer, J., and Fäh, D.: Localization of seismic events produced by avalanches using multiple signal classification, Geophysical Journal International, 216, 201–217, https://doi.org/10.1093/gji/ggy394, 2018b

L127 – non-background noise signals: having 912 non-background noise signals that then split into avalanche and noise events is a confusing terminology. Consider using a different term for non-background noise signals.

We have simplified this formulation by just naming it «events»: In total, we picked 912 events...

**L139 – Fig. 2b: The labels of the panels are missing in Fig. 2.**

We have improved this figure according to another reviewer's comment and therefore adapted the phrasing to:

... middle column in Fig. 2).

L148 – scores exceeded 1.5 : Consider changing to was at least 2.0. Readers just quickly skimming the manuscript might misinterpret it as  $\geq$  1.5

We have changed the sentence accordingly.

**L167 – even: change to roughly even.**

We have changed the sentence to: ...were approximately balanced.

L167 to 169: You differentiate between dry and wet avalanches here but never address it in the results.

With this information, we want to express that we want to develop a model capable of detecting all types of avalanches. Therefore, we must ensure that the training and testing sets contain both types of avalanches.

**L173: Sect. 4.3 referenced before 4.2.**

These references have been changed through the restructuring of section 4. However, we have made sure that the subsections are referenced sequentially.

Fig 3: Could add brackets to the y-label clearly indicating which are training and test folds.

We have improved this figure to be more informative. In this process, we have considered this feedback and labelled the folds explicitly as train or test folds.

Fig 4: Only the encoder is shown. Is the decoder part of the autoencoder not used? Yes, the decoder is discarded during inference. To make this clear, we have added this information to the figure's caption. It is also provided in the main text under the section "Autoencoder features".

**L194: Why is the decoder discarded?**

The decoder is only used during training. Its main purpose is to decompress the features and reconstruct the given input signal. The network is optimized by comparing the input and output with the mean squared error loss. However, during inference, meaning when predicting, the random forests only need the learned features for classification. We have made this clearer (Lines 214 - 216).

**L264: Could include these results as supplementary material.**

We have removed this sentence from the manuscript, as incorporating this early stage of development is outside its scope. Since we used the method from Provost et al., 2017 as

our baseline, we implemented the same classifier on top of the autoencoders for better comparison between the methods.

**Reference:**

Provost, F., Hibert, C., and Malet, J. P.: Automatic classification of endogenous landslide seismicity using the Random Forest supervised classifier, Geophysical Research Letters, 44, 113–120, https://doi.org/10.1002/2016GL070709, 2017.

**L280 – Gini information criterion: Missing citation.**

We have added a reference to the work of Breiman, 2017.

**Reference:**

Breiman, L.: Classification and regression trees, Routledge, https://doi.org/https://doi.org/10.1201/9781315139470, 2017

**Fig. 7: Consider adding a colour bar.**

We believe a colour bar would not add much information since the percentages and number of samples are already shown within each confusion matrix. The colours encode the percentages calculated row-wise and should only provide a fast visual representation.

Sect. 5.1: I recommend integrating Sect. C1 into the main text. Otherwise, it is difficult to interpret the results.

Thank you for the suggestion. However, these classification metrics are widely used and well-known in classification and detection. We assumed that readers of the Geoscientific Model Development journal are familiar with them. For this reason, and to avoid elongating the entire manuscript, we kept them in the appendix. Likewise, we did not introduce convolutional layers or long short-term memory (LSTM) cells used in the temporal autoencoder and referred the reader to the original papers.

A nice overview of frequently used classification metrics is presented in:

Hossin, Mohammad & M.N, Sulaiman. (2015). A Review on Evaluation Metrics for Data Classification Evaluations. International Journal of Data Mining & Knowledge Management Process. 5. 01-11. 10.5121/ijdkp.2015.5201.

**Fig. 8: It is not clear what the x- and y-labels are.**

We have clarified this in the figure's caption.

**Fig. 10 – up/down: Change to top/bottom.**

We have changed it in the revised version.

L331 and 385: Sect. 6.4 referenced before 6.3. I recommend to first discuss the missed avalanche windows and false alarms, and then the applicability to early warning systems.

Thank you for this suggestion. We agree and hence have swapped the sections.

L367: Same as the first sentence of the paragraph.

We have removed the duplicate. Thank you for pointing out.

L378: dotted blue line in Fig. 11.

We have added «...in Fig.11».

L409 to 411: It would be interesting to see how the issue of identifying avalanche onset evolves under different signal window lengths (e.g., trading slightly later detection for overall better performance).

Thank you for this great suggestion. We will consider it for future work.

**L441: What could these unknown sources be?**

Speaking from experience with these models, a source that was often mistaken for avalanches is aeroplanes or helicopters. These air vehicles display the characteristic spindle-shaped waveform of avalanches, but they show higher frequencies in the spectrograms. The other false detections were impossible to identify.

L445: Compared to previous studies, why can the models discussed here better differentiate between avalanches and earthquakes?

The previous studies do not explicitly state that the models had difficulties differentiating between avalanches and earthquakes. Looking at the study of Provost et al., 2017, they did not have problems distinguishing earthquakes from landslides either.

L466: How do the approaches of Bessason et al. (2007) and Hammer et al. (2017) compare the this study?

We did our best to compare our results with previous studies. However, both of these studies do not present metrics to compare, which makes it difficult. In general, in earlier studies, there seems to be no consensus on which metrics to present. This is part of why we reported all metrics, potentially making this work a basis for comparing future implementations and similar studies.

L504: I recommend ending with your key takeaway.

Thank you for the suggestion, we have modified the last paragraph in the conclusions, trying to present the key takeaway (Lines 555 – 565).

I hope the authors find my comments helpful. Sincerely, Kevin Hank

**Referee 6#:**

**General Comments**

The study at hand investigates the possibility of detecting avalanches in seismic sensor signals with the help of representation learning. It demonstrates that an auto-encoder can learn useful features from seismic sensor signals in an unsupervised fashion. The usefulness of those features becomes apparent when training RF classifiers on top of the autoencoder to address binary avalanche detection tasks. Those validation tasks have a high scientific significance and are relevant for operational services in the avalanche warning community.

Scientific significance: The study could contribute significantly to the current field of avalanche detection from seismic sensors. Previous work is limited to random forest approaches and manually engineered features. The authors' approach to learning a representation of the avalanche features and training a classifier on top is a well-respected method in machine learning. While the benefits of this method have not been very well presented in the paper, I acknowledge and recognize the high scientific significance of this work, as well as its novelty.

Scientific quality: The scientific approach is sound and convincing. However, the usefulness of the novel methodological approach is not evident, given that related work outperforms this approach. Potential advantages of the novel method are claimed, but they are neither backed by literature nor experiments. Moreover, the experimental setup must be revised to address relevant concerns regarding the data processing (concerning data split and data normalization).

Scientific reproducibility: The ML part of the work is fully reproducible. The data and the code are accessible and executable. The experimental setup to collect the data is made available and could be reproduced for different site locations. It is unclear how to reproduce the processed data from the raw data.

Presentation quality: The presentation quality could be improved significantly. All requests on that end are "minor," i.e., they involve mainly restructuring bits and pieces, including literature to back up claims, building a more substantial narrative throughout the paper, and editing the paper to respect "context-content-conclusion." Even though my vote is low in that category, I am convinced that the authors can address all those concerns.

Detailed feedback can be found in the referee report.

I am listing here the major revisions requested that I consider the most relevant (emboldened in the feedback document):

**Presentation:**

- Research questions and contributions should be described more explicitly.
- Improve the narrative/storyline of the paper.
- Structure the paragraphs with "context content conclusion" format.
- Use names/identifiers (for models) and technical terms consistently throughout the paper.
- Use booktabs to format the tables.
- Improve figures by adding more descriptive subcaptions. Each figure should bring one main point across. Make the content in the figures more accessible (see comments).
- Discussion: Back up some of the claims on the ML side with literature.

**Methods:**

- Data split: How is data leakage via correlated samples prevented? Are you employing a stratified split? Are you separating sensors/locations?
- Normalization of the data: This is a major revision that is necessary. See the referee report for reasoning.
- Balancing of data: Address this in the paper (explain what you do to address the imbalance).
- Learning curves: Include them in the appendix so the reader can see whether the models are learning well and whether they underfit/overfit.
- More runs: Run the models at least 3 times with different random seeds. If you have already done that, indicate this in the paper and add error bars where applicable. This is necessary to compare the different models. The results are so close that the differences might not be significant if re-run.

**Code and Data:**

- Make sure to store the data on zenodo separately from the code.
- Minor revisions on the code; see referee report.

I recommend the paper for publication after major revisions. I consider the work done here an essential contribution to the field - a step towards automating avalanche detection and warning. Using representation learning in avalanche detection is a major novelty, and the work presented here has the potential (after revisions) to be an outstanding example of using representation learning for a relevant task.

We want to express our gratitude for the thorough and careful review. The feedback was immensely helpful in enhancing the manuscript and strengthening the storyline. We have tried to follow the extensive suggestions while also considering the comments from earlier reviewers. Below, we provide comments and responses to the reviewer's list of suggestions. For detailed implementations and modifications, we refer to the marked-up manuscript.

**Specific Comments**

**Comments in Manuscript**

**Major revisions are emboldened.**

Minor revisions are not emboldened.

**1. General**

- 1. Research questions and contributions should be described more explicitly.
- 2. Add task description (and why those tasks address your research questions)
- 3. **Improve the narrative/storyline** of your paper. Sometimes, explanations or context is missing, and "why" you have made certain decisions is often unclear. You also have very strong arguments at hand from time to time that you are not mentioning or considering.

Thank you for the general suggestions. We have carefully reorganised and improved the introduction section to make the research question clearer (e.g. Lines 72 - 76). In addition, we tried to explicitly describe the different tasks at the end of the introduction (Lines 85 - 87).

Considering the storyline, we have made a considerable effort to improve several sections, as suggested by the reviewer. In particular, we restructured the introduction, the methods and the discussion section. For instance, in the introduction, we have concatenated all traditional approaches to detect mass movements into one paragraph (Lines 46 - 60), highlighted the field of representation learning (Lines 61 - 71) with references to high-impact works and justified the choice of this approach. The «Model development» section was entirely restructured following the reviewer's suggestion to clarify the single steps and decisions (see point 7. below). In the discussion, we have moved the « Applicability to early warning and monitoring systems» section behind the «Missed avalanche windows» and «False Alarms» sections. Moreover, as suggested by Referee #4 we have moved parts of the methods section to the discussion.

Finally, we have aimed to motivate our decisions explicitly by referring to the model tuning process we applied to find all the parameters associated with the models. We also improved the description of the data preprocessing and motivated the chosen normalisation in section 3.3 (Lines 175 - 179). Additionally, the reasoning behind why we have chosen, e.g., to separate the feature extraction and classification, can be found in the discussion.

**2. Structuring**

- Sections sometimes need to be structured better.
   We have restructured most of the introduction and method sections.
   For details, see point 1. or the marked-up latexdif file at the end of this PDF.
- 2. **Context content conclusion:** For most parts, you adhere to that, but several subsections do not follow that structure. Using CCC

consistently would significantly improve the flow and accessibility of your manuscript

We understand the advantages of this structuring and therefore have tried to adhere to it more closely. Nevertheless, where we thought conclusions were not needed or would add duplicate information, we have not. We generally followed the reviewer's marked comments in the manuscript on this concern.

Specifically, we have applied the context-content-conclusion structure to the last paragraph of the introduction, the «Study site and Instrumentation» section, the «Signal windowing, normalisation and dataset splitting» subsection, the «Feature classification» section and all newly created sections.

**3. Language**

1. Consistency: The paper's language is sometimes inconsistent. You often switch between descriptors (sensor array vs multiple-sensors). You do not refer to your different models consistently with the same name. For you, the synonyms are clear, but for the reader, this can be very confusing: It is unclear if two terms are referring to the same concept or (slightly) different concepts and whether that difference is relevant.

Thank you for pointing out this potential confusion. We have carefully reconsidered the naming of different concepts in the manuscript and named them consistently throughout the text. We refer to the methods as a) the baseline for the engineered feature extraction, b) spectral autoencoder or SAE and c) temporal autoencoder or TAE for the learned feature extractors. To remind the reader what we mean by baseline, we specify it now and then, e.g. «...the baseline, i.e. feature engineering...» Moreover, we ensured to always refer to the seismic sensor array as «sensor array» or «array of sensors», as we understand this might be confused with the array as a data structure.

- 2. Explicit is better than implicit (explain things as explicitly as possible). We have followed this suggestion whenever it was appropriate. For instance, introductions of general concepts are sometimes still implicit, but we made all explanations of the implementations explicit.
- 3. Consider running the manuscript through a language correction tool I am not a native speaker, but some formulations sound "off" to me. We have already used language correction software for the last submission and plan to do the same for the revised submission.

**4. Formatting**

Make sure to add links for figure and section labels.
 All links to sections and figures were tested to work in the last submission with a standard PDF reader. We will test all of them again before submitting the revised manuscript. For clarification, LaTeX links the

numbers following the section or figure label, not the label itself (e.g. Sect.  $\mathbf{1}$  or Fig.  $\mathbf{1}$ ).

**2. Make sure all figures are highly resolved.**

We have changed the format of most figures from PNG to PDF, particularly all figures highlighted by the reviewer.

**3. Change tables to booktabs.**

From our point of view, the tables present a complete view of the model performance, which is intended to facilitate comparability for future implementations in this field. Regarding the format, we have already used the LaTeX booktabs package and followed the Copernicus template as required. As we understand the submission and publication process, the manuscript, if accepted, undergoes another stage of formatting and spelling corrections to adhere to the GMD journal standards. We will follow the editor's instructions.

**4. Fix appendix issues.**

Again, we have followed the Copernicus template and trust these issues will be resolved during the potential final production. However, we have tried to reorganise the appendix accordingly.

**5. Data**

1. Label incompleteness: What if experts have missed avalanches (unknown avalanche activity?)

By consulting the imagery from the automatic cameras and the radar detections, we were able to identify all days with avalanche activity. We then analysed them by visually scanning the 24-hour data streams and picking all signals exhibiting a high signal-to-noise ratio. The three experts then labelled the picked signals.

To clarify this, we have reformulated parts of the «Event picking and signal processing» section.

Nevertheless, through the later expert labelling process, some minor avalanches or those flowing at the sensors' detection limit may have been mistakenly assigned to the noise class, i.e. false negatives. These are the potential avalanche signals that received an expert score of 1.5 or lower. However, missing some of these avalanches is inevitable and inherent to the challenging task of detecting avalanches.

2. Label uncertainty: Should the task be considered a regression task instead of a binary classification task?

Thank you for this suggestion. In our setup, we aim to classify a window into clear and distinct classes. We can imagine a setup in which it could be viewed as a regression task using the expert scores. However, in this case, the correctness and completeness of

the label would be even more critical, which is a step our initial implementation could address. We could relate a probability of classification to the expert scores and verify that the classification is, so to say, calibrated. But we leave this for future work.

- 3. Label noise: The agreement score of expert labels is relatively low. Evaluating the performance of the DL model is difficult with that.
- 4. Question: How would you evaluate the success rate of expert labeling? Can you evaluate the ML model independently of expert labels by any chance? Is there a theory on what avalanche signals look like in the Fourier domain, and could you calculate some score based on that? To our best knowledge, the set of data we used to validate and compute generalisation scores is as correct as possible. To this end, we consider this labelled set as accurate as possible, and the ML models are now compared to this gold standard. In a practical implementation, labels won't be available until after an expert assesses a given prediction, which, in this case, we would evaluate the precision of the model (i.e., out of the model's predictions, how many are actual correct events). This would be a straightforward evaluation without requiring experts, but it is surely weaker. The decomposition of signals into the Fourier domain largely depends on the size of avalanches, density/compactness of snow, soil/rock types, water content, and the distance from the avalanche to the recorded sensor. Although there is some theoretical understanding of this aspect, using this knowledge to assess the model's output would still require the latter to be accurate, so we did not venture into this evaluation setup.

**6. Data processing**

- 1. How do you get the data provided in your repository? I assume this is not the raw data from the sensors? Making that bit accessible / describing it would be important to improve reproducibility. The data in the repository are the 10-second windows bandpass-filtered from 1 to 10Hz. To clarify this to users, we added a more descriptive readme to the data directory. The provided data and the code repository let the user reproduce all results. We did not provide the raw data and the accompanying processing code since the main objective of this study is model development, not data acquisition. However, we have decided to upload the raw seismic events with the corresponding labels to a separate Zenodo repository (DOI: 10.5281/zenodo.14892926).
- 2. Extending the dataset: Have you considered extending your dataset with samples from previous publications? (Provost et al. (2017) + Rubin et al. (2012))
  - No, we have not considered it for this study. This is a good point and will surely be a topic to address in the future. To have accurate systems at different geographical locations, we must guarantee that the ML system can generalise to other datasets, even potentially acquired by

different sensors and settings. Once verified, we can assess performance by including several datasets in the training set and studying how the model generalises. This would be a crucial step before applying such a system in practice.

3. Data split: you should also separate locations / sensors, so you do not have leakage via correlated samples. Are you doing a stratified split?
I.e., are the same amount of pos/neg samples available in each split?
Or is it a random split?

We carefully picked three dates to split the dataset to a) prevent leakage between the four folds entirely and b) have roughly even class distributions (see Figure 3). In that sense, we manually defined stratified splits. The correlated samples from the five sensors were always assigned to the same fold, hence not leaking any information. We purposely avoided any look-ahead implementation through appropriate splitting and data normalisation.

4. **Normalization of your data.** Look at feature transformation and data normalization techniques, make your choices, and describe them in the paper.

We developed the presented methods based on 10s seismic signal sub-sequences generated with a windowing algorithm applied to the entire event signals. In general, input normalisation is crucial when training neural networks (Sola and Sevilla, 1997) and even more so when using subsequences of entire time series (Rakthanmanon et al., 2012; Lima and Souza, 2023). In our case, we must avoid implementing a look-ahead normalisation since, at inference, the characteristics, e.g. maximum absolute amplitude, of an event signal are not known in advance. Thus, we normalised each subsequence independently.

Alternatively, normalising all data by any high-level value of the training dataset was not intended. Due to the unsupervised nature of the autoencoders, we were not allowed to differentiate between classes when retrieving such high-level dataset values. Therefore, normalising the entire dataset by the maximum absolute amplitude in the train set, for instance, would have meant normalising avalanches by the maximum amplitude within the noise class. This would have led to input samples differing by order of magnitudes and destabilised training. We made the choice of normalisation explicit by adding it to the «Signal windowing, normalisation and dataset splitting» section.

**References:**

Sola, J. and Sevilla, J.: Importance of input data normalization for the application of neural networks to complex industrial problems, IEEE Transactions on Nuclear Science, 44, 1464–1468, https://doi.org/10.1109/23.589532, 1997.

Rakthanmanon, T., Campana, B., Mueen, A., Batista, G., Westover, B., Zhu, Q., Zakaria, J., and Keogh, E.: Searching and mining trillions of time series subsequences under dynamic time warping, in: Proceedings of the 18th ACM SIGKDD International Conference on Knowledge Discovery and Data Mining, KDD '12, p. 262–270, Association for Computing Machinery, New York, NY, USA, https://doi.org/10.1145/2339530.2339576, 2012.

Lima, F. T. and Souza, V. M.: A Large Comparison of Normalization Methods on Time Series, Big Data Research, 34, 100 407, https://doi.org/https://doi.org/10.1016/j.bdr.2023.100407, 2023.

- 5. My thoughts are to do batch normalization or normalize with sensible physical values (max value recordable via sensor).
  - 1) Generally, data normalization is necessary for your model activation functions and to achieve good and stable learning. We strongly agree, and therefore, we used input normalisation and implemented batch or layer normalisation.
  - 2. 2) You want to focus on the data's pattern, not the scale. I hypothesize that this will improve your model's ability to detect small avalanches or avalanches that are farther away. We want to detect all avalanche sizes, types, and early stages of the avalanche motion characterised by lower amplitudes due to a lower generation of seismic energy and longer source-receiver distance. This is precisely why a normalisation should be applied to the windows independently, as we did. With that, we solely focus on patterns, not the scale.
  - 3. 3) You have different sensors at different geo-locations. Yes, but our objective was to implement a method based on a single sensor. Considering possible future scenarios at various sites, the deployment of sensor arrays is not necessarily guaranteed.
- 6. Imbalanced data: Explain how you address this (via sampling strategy?) As already mentioned in the training procedure and described in Appendix D, we used the weighted random sampler method to address the class imbalance during training. We agree that this was an essential part of the model development. To emphasise this step, we moved and adapted the explanation from the appendix to the main text (Lines 249 257).
- 7. Data windowing: Are you adding additional features from larger window sizes? Such as mean, std, and frequency spectra to provide long-term context?

No, we did not. The methods were derived from 10s seismic signals without considering any longer-term context. We understand and

agree that by reducing the window size, the characteristic spindle shape of avalanche events is lost at some point. However, we found ourselves in a situation where we had to balance the number of training samples with the window size. In earlier studies, similar window sizes were used, so we favoured a larger dataset. Given more avalanche events, we suggest optimising this length in future work.

7. Model section: Please restructure the section. Make clear separations between the feature extractor and the evaluation task. Make it clear that you validate the autoencoders on a separate validation task. Separate our hyperparameter tuning. Etc. I recommend reading a couple of ML papers to get an intuition of how those sections are usually separated and written in a paper.

We have considered this suggestion and made significant revisions to the structure of the model's section:

```
(4) Model development
```

(4.1) Baseline features

(4.2) Autoencoder features

(4.2.1) Architecture

(4.2.2) Training Regime

(4.2.3) Validation

(4.2.4) Model selection

(4.3) Feature classification

(4.3.1) Random forest model

(4.3.2) Cross-validation

(4.3.3) Inference and post-processing

Moreover, we have explained the random forest's inference process in more detail (Lines 311 - 319) and explicitly introduced the prediction post-processing (evaluation tasks) (Lines 320 - 330). Finally, we have described the autoencoder validation in a separate section (Section 4.2.3).

- 8. Model architecture: Please assign speaking names/identifiers to your three models that can be tracked throughout the complete paper (figures, text, abstract, appendix, tables) early on
  - 1. TAE: Why 1d convolutional layers?
  - 2. SAE: Could one use Fourier Neural Operators for this?
  - 3. Have you considered looking into methods to learn from imbalanced data? There are several models out there. Most commonly, people change their sampling strategy for their model to make sure the model sees positive and negative examples equally frequently.

We have followed this suggestion and assigned the model identifiers: baseline, temporal autoencoder (TAE) and spectral autoencoder (SAE).

1. Since we only use vertical components of the sensors and treat each sensor independently, the waveform time series is one-dimensional, i.e. 2000 samples in one channel. Accordingly, we used the frequently used implementation of one-dimensional convolutions for time series (Kiranyaz et al., 2021). To clarify, we have included the survey of

- Kiranyaz et al. (2021) in the manuscript.
- 2. To our understanding, neural Fourier operators are useful ways to learn the embedding of functional spaces, where the learned representation is invariant of the data discretisation and would allow the model to learn complex dynamical systems with long- and short-range dependencies efficiently. Although not impossible, and some properties are indeed appealing, it is not apparent why a model relying on such layers would be performing, particularly because our output is a label space. Hence, the aim is not to learn mapping from dynamics to dynamics. We take this as a suggestion for future work and will spend some more thinking about this. Thank you very much for the suggestion.
- 3. Please see point 6.6. above. We used a weighted random sampler to compensate for the class imbalance by oversampling samples from the avalanche class. To make this critical point in training clearer, we have adapted and moved the section «Appendix D: Weighted random sampler» to the main body of the text.

**References:**

Kiranyaz, S., Avci, O., Abdeljaber, O., Ince, T., Gabbouj, M., and Inman, D. J.: 1D convolutional neural networks and applications: A survey, Mechanical Systems and Signal Processing, 151, 107 398, https://doi.org/https://doi.org/10.1016/j.ymssp.2020.107398, 2021

9. Model learning: Include learning curves and show that the models are actually learning well. The reader wants to know if you are overfitting/underfitting. These figures can be included in the appendix, but they are essential to strengthening your paper. If your models are underfitting or not learning incredibly well, this would be a strong indicator that you can achieve even better results.

We have included the learning curves of both autoencoders in the appendix «E1 Learning curves».

10. Model comparison: **Have more runs** (change random seed) to show whether differences are relevant/significant. Include error bars in your results. If you

already run models across multiple seeds, report the number of runs and error bars.

Thank you very much for your suggestion. We carefully developed the models so as not to be biased towards a specific type of avalanche (wet/dry) or size. Particularly, we made sure the independent test set contained all types and sizes of avalanches and, therefore, was representative of our test site. Additionally, we prevented any data leakage between the data folds by selecting and splitting by specific dates. Considering the reviewer's concern, we understand that having multiple runs is beneficial. Therefore, we have retrained them 20 times with varying seeds (powers of two starting at  $2^0$ ). Similarly, we have retrained the random forest model of the fixed baseline features with these seeds and the random forest models of the autoencoder features with these seeds on the correspondingly extracted seeded features. The following figure and table compare the models across these runs. We observed consistent performance and stability with avalanche f1-score variability being at most 1% in all three methods.

As we have considered this suggestion particularly valuable, we have adapted all figures and metrics in the tables as well as the text showing the mean value and the standard deviation.

**11. Figures**

- Each figure should bring one main point / finding across. The current figures show results, but they are not all structured or designed in a way that makes results easily accessible. A reader should be able to read a paper and access all main findings by reading the figures + captions.
- 2. Some figures need more descriptive captions.

Thank you for this mindful suggestion. We have changed the format of the figures to PDF to ensure they are highly resolved when zoomed in. Moreover, we have considered the reviewer's comments in the manuscript and adapted the figures accordingly. In particular, we have improved the figures related to the

methods (Figure 4). Additionally, we have considered all comments in the script regarding the captions and written them more descriptively.

**12. Discussion**

1. You need to back some of your claims (scalability, ability to generalise, etc.) with literature.

We agree that these claims need to be proven first. Therefore, we have removed or modified them and referred to future work investigating these points.

**13. Conclusion**

- 1. Adapt the sentence "We have shown that it bears strong potential..." (see comments). You have shown it can keep up with current state-of-the-art avalanche detection methods. Still, you have not demonstrated its potential (scalability, generalisation ability, etc) in your experiments.
- 2. A more memorable last sentence talk about downstream applications (operational! Avalanche warning! You have such a strong and relevant use case, and your audience wants to hear about that.) would strengthen your conclusion.

Thank you for these suggestions. We have modified the last paragraph of the conclusions accordingly (Lines 555-565).

**14. Code**

- 1. Looks good! Clean repo, nicely coded, well done.
- 2. You could add more documentation.
- 3. You have the code two times in the repo one seems to be for MacOS? Is this on purpose?

No, this was not on purpose. We have removed these files and all «DS\_Store» files.

4. Mention the Python version in the Readme.

We have used the Python version 3.9.7 and have added it to the Readme.

5. Add versions of your packages in requirements. Your code will not be reproducible later on if you do not report that (and it is not maintained).

We have included all package versions in the requirements.

6. Dir "models" and "lib" should be in "code" (semantically). I also get import errors if I do not move them over there.

We have tested the code again, and indeed, it showed import errors. Thank you for noting. One workaround was to change the PYTHONPATH variable, i.e. «export PYTHONPATH=\${PYTHONPATH}:\${HOME}». To avoid this, we have moved all files from inside the «code/» directory to the parent directory.

- Using your panda version broke the code for me (numpy –
  pandas incompatability). I upgraded pandas to fix it.
  This has been resolved by specifying the package
  versions (see Point 5).
- 8. Consider using Pathlib to handle path os-independently.

  Thank you for the hint. We have modified all path-related code lines to use the package Pathlib and successfully tested all functionalities again.
- 9. Add in your readme where people must change paths to get your code running on their machine.

This information has already been part of the Readme but might have been misleading. We hope to have clarified it by using Pathlib and adding the following to the Readme:

Upon successful installation the root directory needs to be changed in lib/utils/variables.py!

ROOT\_DIRECTORY = Path.home() / 'path' / 'to' / 'project'

**15. Data**

Make sure to store the data on zenodo separately from the code.
 You will want to update the data without updating the code in the
 future. If you want the data to be accessible (for other researchers
 and their projects), it needs to be maintained separately and with its
 own version control.

We understand this point and also believe research should be open to everyone. Therefore, we have created a new data repository where the raw seismic data of all events used in this study and their corresponding labels are available (DOI: 10.5281/zenodo.14892926). In the future, we can update this repository with the latest events recorded at our test site. Nonetheless, we have retained the processed 10-second windows in the code repository to maintain conciseness and facilitate the models' downloading and testing. We hope to serve the research community with this.

---

## Author Response (AR3)

Dear editors, Dear reviewers,

We highly appreciate that the referees have taken the time to review the revised manuscript once again. Their reviews and suggestions have contributed immensely to improving the manuscript and the content. For this, we want to thank them sincerely.

Kind regards, Andri Simeon

**Table of Contents**

| REFEREE 3#:       | 2 |
|-------------------|---|
| GENERAL COMMENTS  | 2 |
| REFEREE 4#:       |   |
| GENERAL COMMENTS  |   |
| SPECIFIC COMMENTS |   |
| REFEREE 6#:       | 5 |
| CENEDAL COMMENTS  |   |

**Referee 3#:**

**General Comments**

I found the manuscript easier to read than the previous version (but probably because I already spent a lot of time on the previous version).

Most questions have been addressed, thanks.

But it is a bit frustrating having to wait for "future work" to test several ways to improve the method (including network and polarisation features, infrasound sensor, longer time windows...).

Thank you very much for your helpful feedback and for taking the time to review our manuscript carefully. We truly appreciate your suggestions regarding possible improvements to the method. At this stage, however, the paper is finalized and already large. We agree that exploring aspects such as network and polarisation features, infrasound sensors, and longer time windows would be highly valuable, and we will address these promising directions for future research.

Just one new question. Did you only apply your model to time periods when experts manually picked and classified events, or did you (ou could you?) apply your model on the full dataset, including summer periods? This could be interesting to further estimate the rate of false avalanche detections (eg, when there is no snow) and to compare the rate of avalanche activity with the estimated avalanche hazard.

Yes, so far we have only tested the models on the manually picked events. A large-scale transfer of these models to continuous data will certainly be considered for future work. However, we expect this transfer to be non-trivial and will likely involve further model development and tuning.

Due to restrictions from landowners, the sensor array can only be deployed during winter. Therefore, the dataset does not contain days in summer.

Can you explain in the zenodo archive what is shown in the catalog labels.csv: what is the label? av\_score? eq\_score?

We added a description to the repository to explain the columns in the labels.csv file.

By curiosity, is the seismic data also available from the FDSN?

Currently, it is only available on the Zenodo repository. Since the sensors are not running in summer, and the test site is not permanent and bound to the project duration we did not consider FDSN.

**Referee 4#:**

**General Comments**

This manuscript develops autoencoder derived seismic attributes and engineered seismic attributes as features in a random forest classification detection of snow avalanches. The results suggest that the autoencoder derived attributes perform as well as the engineered seismic attributes for event detection. The avalanche detection method is reported to be potentially used as an operational, near real-time avalanche detection system, though the relatively high number of false alarms requires further improvement. The work is presented with respect to the previous studies employing machine learning for seismic event detection while highlighting their significant and novel contribution of unsupervised feature extraction. I found the paper to be informative and complete in analysis and have been satisfied by the authors' responses to the initial review which improved the methodological development and comprehensibility of the work.

We are very grateful for your careful review and are happy that we followed the previous reviews satisfactorily.

**Specific Comments**

Line 6: "Therefore, we compiled a dataset of seismograms recorded with an array of five seismometers..." This sentence seems a bit disconnected from the main ideas presented. Is this statement intended to link back to "Monitoring snow avalanche activity is essential for operational avalanche forecasting..." or "Still, automatically distinguishing avalanche signals from other sources in seismic data remains challenging." I think the intent could be clarified by replacing "Therefore" with a descriptive intro to sentence like, "Because of the inherent complexity of interpreting signals travelling within the subsurface, we utilised an array of five seismometers..." This example expresses the importance of having an array of seismometers.

This sentence was intended to refer back to the challenge of distinguishing an avalanche from a noise signal, for which we needed a comprehensive dataset of avalanche and noise samples. Following your suggestion and to make it clearer, we changed the sentence to: «To study and interpret the variety of these signals, ...»

Line 104 – 106: Feels like a run-on sentence. Consider the revision, "Additionally, the site was equipped with a Doppler radar and three automatic cameras to obtain independent validation data, including accurate release times and information on the type and size of avalanches, provided favorable weather conditions."

Thank you for the suggestion. We changed this sentence accordingly.

Line 109- 110: "The cameras automatically photographed all surrounding slopes every 30 minutes (Fig. 1)." Consider including one sentence detailing how the photographs were utilized. Manually inspected as a corroboratory inspection of radar or other data

source detections, or automatically reviewed as an independent method? This is mentioned on lines 113-114, but not explicitly.

To clarify this, we added: «..., which we manually inspected to identify days with avalanche activity and verify avalanche events of the detection systems.»

Line 132: What is exactly meant by ground velocity? The derivative of the seismic displacement? Just a bit more detail on this method would be helpful.

We changed this sentence to: «..., we transformed the units of the raw recordings, i.e. counts, to meters per second (ground motion)». Seismometers detect ground vibrations and convert them into an electrical signal. The data logger then digitizes this signal and stores it as counts. To convert raw counts into physical units, the recorded counts are divided by the seismometer's sensitivity factor, yielding ground velocity in meters per second.

Line 232: "As activation function -> As an activation function"

Line 282: "Therefore, we used the three train folds -> Therefore, we used the three training folds"

Line 306: Similarly "train" -> "training"

Line 307: "weigh" -> "weight"

Line 465 "Tough" -> "Though"

We adapted all of the above five suggestions accordingly.

One thing to note: Between the Engineered Feature and AE feature avalanche detection, which had comparable recall values, were the same avalanches detected? Could the implementation of both SAE and engineered feature detection further increase the detection capability. For if they detected the same amount of avalanches, but different ones, perhaps this increases the overall detection. This analysis is explicitly missing, but could provide additional insights to the differences of the detection methods. Perhaps this is something you have already investigated, but did not note in the manuscript. Figure 13 touches on this conceptually, but it is still hard to discern if the false positive detections stem from different events or not.

This is indeed an interesting question. Unfortunately, we observe that the presented models strongly agree on the separate avalanches, meaning they detect and miss the same avalanches. Therefore, combining the models would lead to similar or even worse results. The reason for the agreement of the models is instead found in the respective signals. We qualitatively observed that true detections stem from strong avalanche signals, i.e. relatively large avalanches or avalanches flowing in the proximity of the sensor array. In contrast, weaker avalanche signals are often missed by all methods, which is what we had expected.

With Regards, Tate Meehan

**Referee 6#:**

**General Comments**

I am accepting the manuscript "as is". The updated manuscript addresses all my concerns brought forward in my last review. Specifically, my ranking of the scientific quality and presentation quality has increased significantly. The scientific significance and reproducibility were already present during the last iteration (and still are).

I want to highlight the great effort the authors made to rework a large part of the manuscript, fixing code issues, updating figures, and even rerunning the experiments on a larger scale. All major and smaller suggestions have been taken into account and have led to either an adaptation of the manuscript or have been backed up by the authors with detailed explanations. Algorithm design choices were taken carefully, and are now communicated more clearly to the audience.

I thank the authors for their patience with the review process and for contributing their work to the scientific community.

We want to thank you for the careful past review. It has helped us immensely in improving the manuscript. We are happy that we could meet the previous concerns.